# Risk of punishment influences discrete and coordinated encoding of reward-guided actions by prefrontal cortex and VTA neurons

Junchol Park[1†], Bita Moghaddam[2*]

[1]Department of Neuroscience, University of Pittsburgh, Pittsburgh, United States;
[2]Department of Behavioral Neuroscience, Oregon Health and Science University, Portland, United States

**Abstract** Actions motivated by rewards are often associated with risk of punishment. Little is known about the neural representation of punishment risk during reward-seeking behavior. We modeled this circumstance in rats by designing a task where actions were consistently rewarded but probabilistically punished. Spike activity and local field potentials were recorded during task performance simultaneously from VTA and mPFC, two reciprocally connected regions implicated in reward-seeking and aversive behaviors. At the single unit level, we found that ensembles of putative dopamine and non-dopamine VTA neurons and mPFC neurons encode the relationship between action and punishment. At the network level, we found that coherent theta oscillations synchronize VTA and mPFC in a bottom-up direction, effectively phase-modulating the neuronal spike activity in the two regions during punishment-free actions. This synchrony declined as a function of punishment probability, suggesting that during reward-seeking actions, risk of punishment diminishes VTA-driven neural synchrony between the two regions.
DOI: https://doi.org/10.7554/eLife.30056.001

*For correspondence:
bita@ohsu.edu

Present address: †Janelia Research Campus, Howard Hughes Medical Institute, Ashburn, United States

Competing interests: The authors declare that no competing interests exist.

## Introduction

Goal-directed actions aimed at obtaining a reward often involve exposure to an aversive event or punishment. For example, foraging for food in the wild may result in encountering a predator. In a causally and socially complex world, appropriate representation of punishment that lurks around during reward-seeking actions is critical for survival and optimal action selection. Deficits in this representation may be associated with detrimental behavioral patterns observed in impulsive behavior and addictive disorders while exaggerated representation of punishment risk may be linked to anxiety disorders (*Bechara et al., 2000*; *Gillan et al., 2016*; *Hartley and Phelps, 2012*; *Lee, 2013*; *Mineka et al., 1998*).

How is risk of punishment associated with reward-seeking actions represented by the brain? To begin to address this question, we focused on the ventral tegmental area (VTA) and the medial prefrontal cortex (mPFC). Neurons in the VTA including dopamine (DA) and non-dopamine neurons are critical components of the reward circuitry including reward-mediated actions (*Cohen et al., 2012*; *Matsumoto et al., 2016*; *Roesch et al., 2007*; *Schultz, 1998*; *Tan et al., 2012*; *van Zessen et al., 2012*; *Wise, 2004*; *Wood et al., 2017*). We, therefore, hypothesized that ensembles of VTA neurons represent risk of punishment associated with reward-guided behavior. Importance of VTA notwithstanding, the mPFC is also implicated in reward representation and reward-guided action selection (*Barraclough et al., 2004*; *Buschman et al., 2012*; *Kobayashi et al., 2006*; *Powell and Redish, 2016*; *Rich and Shapiro, 2009*), as well as control of aversive and anxiety-like behavior

**eLife digest** When deciding what to do, we usually try to predict the likely outcomes of our actions. This helps us choose behaviors that will lead to positive outcomes, or rewards, and avoid those that will lead to negative outcomes, or punishments. But in practice, actions that offer the possibility of reward often involve varying degrees of risk. When animals forage for food, for example, they risk encountering a predator. In our complex social world, applying for a job or asking someone out on a date means risking rejection. Being able to weigh up the likelihood of positive and negative outcomes is vital for effective decision-making.

Park and Moghaddam have now studied how the brain's reward system takes account of possible negative outcomes. Rats learned to poke their noses into a window inside a testing box whenever a light came on, to earn a sugar reward. The rats completed three blocks of trials. During the first block, they received only rewards. But for a few trials during the second and third blocks, they also received a mild electric shock as well as their reward.

Throughout the task, Park and Moghaddam monitored the activity of two regions of the brain that encode rewards, the ventral tegmental area (or VTA for short) and the medial prefrontal cortex. The VTA sits deep within the brain and produces the brain's reward chemical, dopamine. The prefrontal cortex is at the front of the brain and helps support cognition. In the reward-only block of trials, neurons in the VTA synchronized their firing with neurons in the prefrontal cortex. In blocks two and three, where there was a risk of shock, this synchrony decreased. This suggests that the prefrontal cortex takes greater control of decision-making when an unpleasant outcome is possible.

Consistently overestimating the risk of things going wrong will lead to anxiety. Underestimating the risks will lead to impulsivity and poor decision-making. Park and Moghaddam's experiment offers a way to study the mechanisms underlying these processes in animals. The results also suggest that using scalp electrodes to track prefrontal cortex activity in patients could be helpful in clinical trials for anxiety or impulse-control disorders.

DOI: https://doi.org/10.7554/eLife.30056.002

(*Adhikari et al., 2010*; *Kim et al., 2017*; *Kumar et al., 2014*; *Likhtik et al., 2014*; *Park et al., 2016*; *Ye et al., 2016*). At the network level, the mPFC and VTA (both DA and non-DA neurons) send reciprocal projections to each other (*Berger et al., 1976*; *Carr and Sesack, 2000a*; *2000b*). The VTA neurons projecting to mPFC have been shown to respond to stressful and anxiogenic drugs with a greater degree of sensitivity compared to the mesolimbic or mesostriatal projections (*Abercrombie et al., 1989*; *Bradberry et al., 1991*; *Moghaddam et al., 1990*; *Thierry et al., 1976*). Furthermore, photostimulation of the VTA DA input to the mPFC elicits anxiety-like behavior (*Gunaydin et al., 2014*; *Lammel et al., 2012*), suggesting a more causal role for this neural circuit in aversive behavior. Given this, we further hypothesized that the interaction between VTA and mPFC provides a dynamic representation of punishment risk during reward-seeking actions.

To test these hypotheses, we first designed and validated a task that allowed us to assess reward-guided actions in the absence or presence of punishment risk in the same recording session. The latter criterion was critical because it allowed us to track the activity of the same ensembles of neurons as a function of punishment risk. The task was designed so that an instrumental action always procured a reward, but the same action probabilistically led to punishment with blockwise varying degrees of contingency. Thus, different blocks had varying risk of punishment associated with an action while action-reward contingency remained constant. We then recorded single unit activity and local field potentials (LFP) from the VTA and mPFC simultaneously during this task. The simultaneous recording allowed us to compare the inter- and intra-regional neural codes for punishment risk-based modulation of behavioral events.

At the single unit level, we found that VTA and mPFC neurons encode risk of punishment and blockwise/trial-by-trial behavioral modulation during action execution. At the network level, we found that coherent theta oscillations synchronize the VTA and mPFC in a bottom-up direction, effectively phase-modulating the neuronal spike activity in the two regions during punishment-free actions. This oscillation-mediated neural synchrony declined as a function of punishment risk, suggesting that desynchronization of coordinated activity in these two regions signals punishment.

## Results

### Impact of punishment risk on reward-seeking behavior

Blockwise changes in punishment risk resulted in changes in behavior (*Figure 1a–b*, *Video 1 and 2*). Videos are representative examples of trials in block 1 (*Video 1*) and block 3 (*Video 2*) of the same animal. As clearly captured in *Video 2*, animals do not merely freeze during RTs; they make incomplete nose pokes, which, importantly, are selectively observed during instrumental pokes but not during trough pokes to retrieve reward. The mean response time (RT) and time spent in immobility during RT markedly increased as a function of punishment (*Figure 1c*; GLM repeated measures, $F_{2, 32}$ = 24.94, p<0.001; GLM repeated measures, $F_{2, 32}$ = 22.44, p<0.001, respectively). These increases reflect a state anxiety but not diminished motivation to obtain reward as the latency for reward retrieval (reward RT) remained similar across blocks regardless of punishment (*Figure 1c* inset; GLM repeated measures, $F_{2, 32}$ = 2.97, p=0.07). Punishment risk significantly increased the trial-to-trial variability in RT (*Figure 1—figure supplement 1a*; GLM repeated measures, $F_{2, 32}$ = 13.89, p<0.001). This increase was associated with shock occurrences, since the RT varied as a function of the trial lag (distance) from the previous shock (*Figure 1—figure supplement 1b*) indicating that animals adjusted their behavior across trials even within the same block in the face of punishment risk.

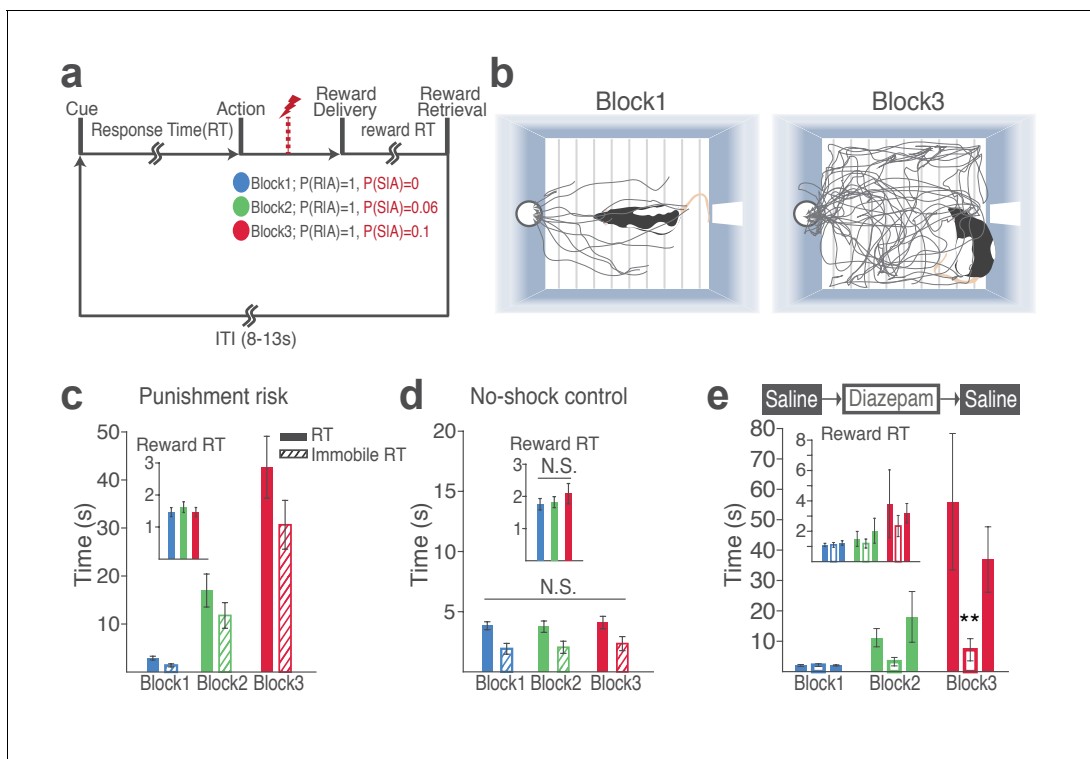

**Figure 1.** Punishment risk induces anxiety-like changes in reward-seeking behavior. (a) A schematic diagram illustrating the task. Punishment risk varied across blocks – Block1, P(S|A)=0; Block2, P(S|A)=0.06; Block3, P(S|A)=0.1. Every nose poke procured reward across all blocks – P(R|A)=1. (b) Representative behavioral trajectories in block 1 (left, 10 trials) and block 3 (right, 10 trials). (c) Significant increases in response time (RT, filled bars, mean ±s.e.m.) and immobile RT (slashed bars – time elapsed motionless during RT) were observed as a function of punishment risk. (Inset) Latency from reward delivery to retrieval (reward RT) did not differ across blocks. 17 rats performed this task with electrophysiological recording. (d) RT, immobile RT, and reward RT did not change across blocks in the absence of punishment (no-shock control session). 12 out of the 17 rats performed this task with electrophysiological recording before they ever received an electrical foot shock. (e) A separate group of rats (N = 9) performed three sessions of the task with pretreatment of saline (Day 1) – diazepam (2 mg/kg) – saline (Day 2). Pretreatment of an anxiolytic diazepam (2 mg/kg) but not saline injection averted punishment-induced increase in the mean RT. **p<0.005; post hoc test. (Inset) Injections did not influence reward RT.

DOI: https://doi.org/10.7554/eLife.30056.003

The following figure supplement is available for figure 1:

**Figure supplement 1.** Punishment risk increases trial-to-trial variability in RT.

DOI: https://doi.org/10.7554/eLife.30056.004

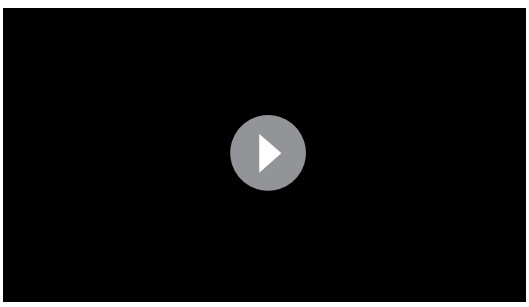

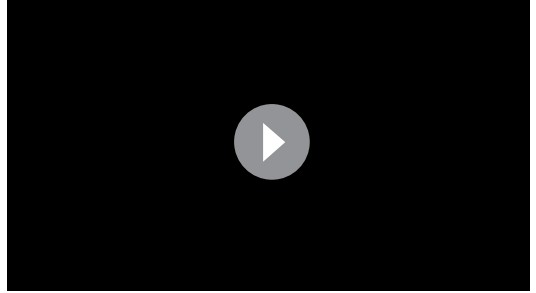

**Video 1.** Example task performance (three trials) in the absence of punishment risk (block 1). Task events are captioned.
DOI: https://doi.org/10.7554/eLife.30056.005

**Video 2.** Example task performance (one trial) in the presence of punishment risk (block 3). Task events are captioned.
DOI: https://doi.org/10.7554/eLife.30056.006

To examine whether the increase in RT reflected learning of action-punishment contingency or merely an immediate impact of shock, we compared RTs of early trials in block 2 and 3, before the delivery of the 1st shock. Rats displayed a subtle but significant increase in RT in block 2, and robust increase in block 3 before the first shock (*Figure 1—figure supplement 1c*; GLM repeated measures, post hoc pairwise comparisons, p<0.001), suggesting that there is learning of action-punishment contingency from previous experiences. During the no-shock control session, when animals performed the same number of trials and blocks in the absence of punishment, the mean RT and the variance RT did not differ across blocks (*Figure 1d*, *Figure 1—figure supplement 1d*).

The punishment-induced increase in RT suggested a transient anxiety-like state during block 3. This was confirmed by the anxiolytic drug, diazepam (2 mg/kg), significantly attenuating the increase in RT of block 3, compared with saline-pretreatment data (*Figure 1e*; Repeated measures ANOVA, $F_{4,\ 48}$ = 3.27, p=0.019; *post hoc test*, block 3, *p* values < 0.01). Diazepam or saline injected animals showed equivalent levels of reward RT (*Figure 1e* inset; Repeated measures ANOVA, $F_{4,\ 48}$ = 0.34, p=0.852; *post hoc test*, *p* values > 0.51).

## mPFC and VTA single units represent punishment risk

During task performance, 167 mPFC and 102 VTA single units were recorded from histologically verified electrodes (*Figure 2—figure supplement 1*). For all single unit data analyses, we classified VTA units into putative dopamine (DA, n = 55) and putative non-dopamine (non-DA, n = 47) subtypes (*Figure 2—figure supplement 2*, Materials and methods). We first examined the trial-averaged neuronal activity of mPFC, VTA putative DA, and non-DA units to compare their general tuning properties during task events – cue onset, action, and reward delivery. *Figure 2a* shows peri-event neuronal activity averaged across all trials and blocks. The majority of VTA putative DA units displayed phasic excitatory responses at each task event as has been previously reported (*Schultz et al., 1993*), whereas non-DA and mPFC units showed weaker and temporally diffuse responses (*Figure 2a–c*; Repeated measures ANOVA, post-cue, $F_{2,\ 266}$ = 22.05; peri-action, $F_{2,\ 266}$ = 43.78; post-reward, $F_{2,\ 266}$ = 48.93, *p* values < 0.001).

We then examined modulation of single neuronal activity across blocks (*Figure 2b–c*). Some of the mPFC, VTA putative DA, and non-DA single units appeared to have modulated their peri-event firing rates across blocks as a function of punishment risk. Given this, we quantified individual neuronal representation of punishment using a percent explained variance (ωPEV) statistic (*Buschman et al., 2012*). This analysis essentially quantifies how much of total variance in a neuron's firing rate across trials can be explained by blockwise changes in punishment risk (Materials and methods). Comparing the ωPEV statistics of the original spike trains with the ωPEV distribution of surrogate spike trains created by shuffling block labels (*Figure 2d–e*, Materials and methods) allowed us to identify punishment encoding units, i.e. units that responded to the same events (cue, action, reward) differently in different blocks. The most robust punishment encoding occurred during the peri-action epoch (*Figure 3*, *Figure 3—figure supplement 1*) given that ωPEVs during this period was greater than that of the peri-cue or peri-reward epochs (*Figure 3a*). There was a significant interaction between time and unit groups in the peri-action ωPEV (*Figure 3a*; Repeated

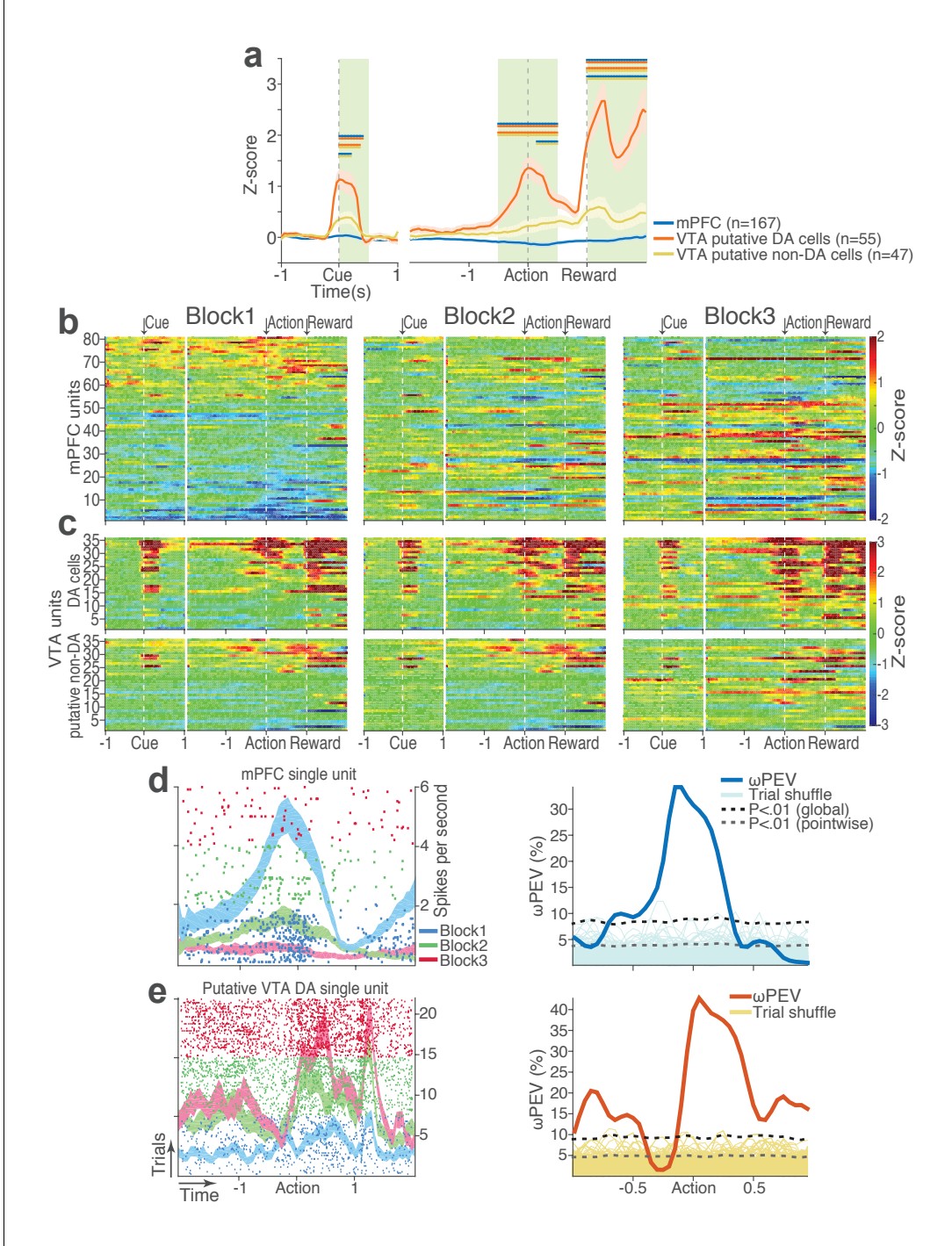

**Figure 2.** mPFC, VTA putative DA, and non-DA single units respond to task events and punishment. (**a**) Peri-event activity averaged across all trials and all units within each neuron group. Dual-colored bars above indicate significant pairwise differences at corresponding time bins according to the post hoc analysis (p<0.05). The green shadows indicate time windows of statistical analyses. (**b**) Baseline-normalized peri-event firing rates of mPFC units are plotted per block to reveal neuronal responses to punishment. Only units with significant activity modulation are plotted – that is, punishment-encoding units (*Figure 3—figure supplement 1*). (**c**) Peri-event activity of VTA putative DA (top panels) and non-DA (bottom panels) punishment-encoding units. (**d-e**) Identification of single units discriminating their firing rates across different blocks as a function of punishment. (**d**) Left, A raster plot showing a representative mPFC unit's peri-action spike activity across blocks with spike density functions of different blocks superimposed. Right, To quantify each unit's encoding, percent variance in the unit's firing rate explained by blockwise change in punishment contingency (ωPEV) was calculated. To determine the global ωPEV band, trial-shuffled surrogate ωPEV distribution (light blue curves) was acquired, and the pointwise and global ωPEV bands were found from the distribution at α = 0.01 (Materials and methods). A unit whose ωPEV curve crosses the global band was determined as a

*Figure 2 continued on next page*

*Figure 2 continued*

punishment-encoding unit. (e) Left, A representative VTA unit's peri-action activity across blocks. Right, This VTA unit satisfied the punishment-encoding criterion.

DOI: https://doi.org/10.7554/eLife.30056.007

The following figure supplements are available for figure 2:

**Figure supplement 1.** Histologically verified placements of mPFC and VTA electrodes.

DOI: https://doi.org/10.7554/eLife.30056.008

**Figure supplement 2.** Classification of VTA single units to putative DA or non-DA units.

DOI: https://doi.org/10.7554/eLife.30056.009

measures ANOVA, $F_{38, 5054} = 2.68$, p<0.001), indicating distinct time-varying patterns in representation of punishment by different unit groups. Similar time-varying patterns were observed in the proportion of units representing punishment risk (*Figure 3b*). To examine the time course of individual neuronal encoding in the peri-action epoch, we recalculated the peri-action ωPEV using a narrower moving window (50 ms width, 5 ms step) to reveal time points of individual neuronal representation of punishment at a higher temporal resolution. Response of VTA putative DA neurons appeared to be concentrated specifically around the time of the action, whereas non-DA and mPFC units displayed temporally diffuse patterns (*Figure 3c*). This result suggested that DA units may be involved in more precise signaling of action and punishment risk, whereas the mPFC and VTA non-DA units may represent more persistent effects of punishment on motivational/emotional states. Consistent with this function, substantial proportions of mPFC (31%) and non-DA (32%) units significantly modulated their baseline firing rates across blocks, suggesting that they represent punishment on a longer temporal scale (*Figure 3d*). Fewer VTA putative DA units (14%) modulated their baseline activity (*Figure 3d*).

Next we examined the direction of neuronal response (excitation or inhibition) as a function of punishment. Similar proportions of mPFC units encoded punishment with bidirectional modulation of activity (*Figure 4a*), which resulted in the lack of net excitation or suppression of activity across blocks (*Figure 4b*; Repeated measures ANOVA, $F_{2, 241} = 2.09$, p=0.126). The majority of VTA putative DA units encoded punishment risk with an excitatory response (*Figure 4c*) specifically at the time of the action (*Figure 4d*; Repeated measures ANOVA, $F_{2, 105} = 3.96$, p=0.022), but not during other task events. Similarly, a greater number of non-DA punishment-encoding units displayed an excitatory modulation of activity (*Figure 4e*) with a trend toward a net excitatory effect of the block (*Figure 4f*; Repeated measures ANOVA, $F_{2, 105} = 2.94$, p=0.057).

To distinguish neuronal encoding of punishment risk from other confounding factors that may cause block-dependent changes in neuronal activity (e.g., satiety, fatigue, or spontaneous drifting over time), we performed a control experiment recording from 126 mPFC and 57 VTA single units (putative DA, n = 28; putative non-DA, n = 29) during performance of three consecutive punishment-free blocks. We observed a much weaker blockwise changes in firing rate indicating negligible impact of confounding factors on the neuronal encoding of punishment (*Figure 5*).

Taken together, these data indicate that both VTA and mPFC single neurons convey information about punishment risk during reward-seeking behavior. Moreover, we observed distinct temporal and directional tuning properties from mPFC, VTA putative DA, and non-DA neuronal subpopulations. This heterogeneity may enable the VTA-mPFC circuit to represent diverse motivational/emotional aspects of punishment risk.

## Punishment risk-induced trial-by-trial changes in response time is tracked by mPFC and VTA neurons

Our behavioral data indicated remarkable trial-to-trial variability of RT in the face of punishment risk (*Figure 1—figure supplement 1*). To better understand the relationship between the behavioral and neural data, we examined whether, and to what extent, RT variability is related to mPFC and VTA neuronal activity. At the single neuron level, only a small fraction of single units exhibited significant correlation between firing rate and the RT across trials (mPFC, 10.2%; VTA putative DA, 36.4%; VTA putative non-DA, 25.5%). This result indicated that inferring trial-by-trial behavior from single neuronal activity was limited. We, therefore, moved on to correlating single-trial RT with neuronal

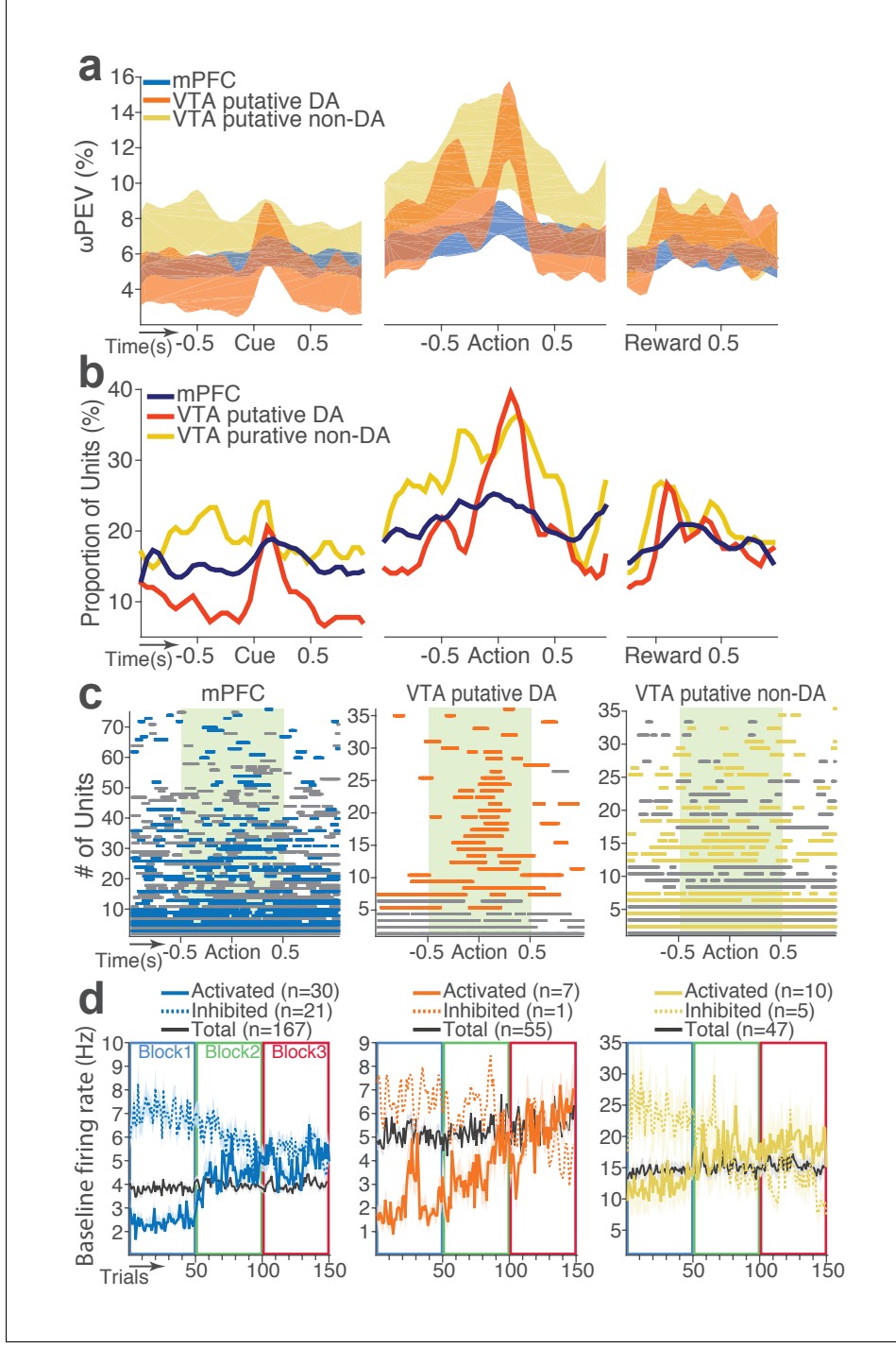

**Figure 3.** mPFC, VTA putative DA, and non-DA single units encode punishment risk by modulating their peri-event and baseline firing rates. (a) Shaded areas indicate the mean ±s.e.m. ωPEV averaged across all units in each neuron group across time. (b) Line plots indicate proportions of punishment-encoding units. (c) To reveal time points of punishment encoding, crossing of the global ωPEV band by each unit is marked with a line segment (Materials and methods). Only the units with at least one crossing are included in each plot. Single units with significant change in their baseline firing rate are marked with gray lines (see below). (d) Subpopulations of single units represented punishment with significant excitatory or inhibitory modulation of their baseline (inter-trial interval) activity. The mean ±s.e.m. baseline firing rates are plotted across trials and blocks.
DOI: https://doi.org/10.7554/eLife.30056.010

The following figure supplement is available for figure 3:

*Figure 3 continued on next page*

*Figure 3 continued*

**Figure supplement 1.** Representative punishment-encoding mPFC (**a-b**), VTA putative DA (**c-d**), and non-DA (**e-f**) single units.

DOI: https://doi.org/10.7554/eLife.30056.011

population activity. A conventional measure of the neuronal population activity is averaging across neurons, but the bi-directionality in the individual neuronal representation of risk (*Figure 4*) undermined the feasibility of this approach. As an alternative approach, we used a neuronal population state space analysis that detected patterns of activity co-modulation in simultaneously recorded mPFC or VTA populations (comprising of 10 or more units). Trial-by-trial neural population trajectories were extracted within this state space using a dimensionality reduction method Gaussian process factor analysis (GPFA; Materials and methods) (*Yu et al., 2009*). Examining the correlation between the neural population trajectory and the RT across trials revealed that mPFC and VTA population trajectories track trial-to-trial variation in RT (*Figure 6*). Specifically, the deviation (Euclidean distance) of each trial's neural trajectory from the block 1 mean trajectory was significantly correlated with the deviation of each trial's RT from the block 1 mean RT. Simply put, the further each trial's neural trajectory departed from the block 1 mean trajectory, the greater the RT appeared (*Figure 6*,

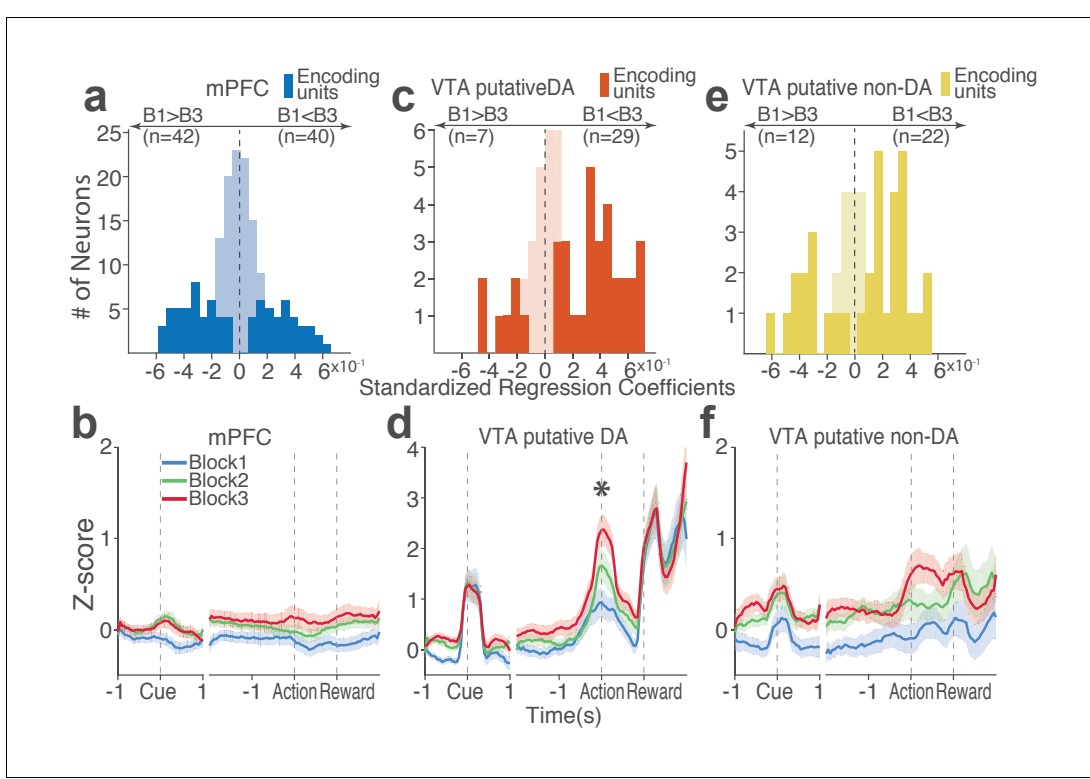

**Figure 4.** Distinct subpopulations of single units represent punishment risk with excitatory or inhibitory activity modulation. (**a, c, e**) Units are distributed horizontally based on modulation of their peri-action activity across blocks as a function of punishment. Standardized regression coefficients (SRC) were computed for a normalized quantification of each unit's peri-action activity modulation by punishment (Materials and methods). In each distribution, units with excitatory or inhibitory activity modulation are located in the right or left portion of the distribution. Punishment-encoding units are solid-colored, while non-encoding units are pale-colored. (**a**) Direction of the mPFC neuronal activity modulation. (**b**) The baseline-normalized activity of the mPFC encoding units per block (mean ±s.e.m.). (**c**) Direction of the VTA putative DA neuronal activity modulation. (**d**) The activity of the VTA DA encoding units per block (mean ±s.e.m.). Asterisk indicates a significant effect of block on the peri-action activity ($p < 0.05$). (**e**) Direction of the VTA putative non-DA neuronal activity modulation. (**f**) The activity of the VTA non-DA encoding units per block (mean ±s.e.m.).

DOI: https://doi.org/10.7554/eLife.30056.012

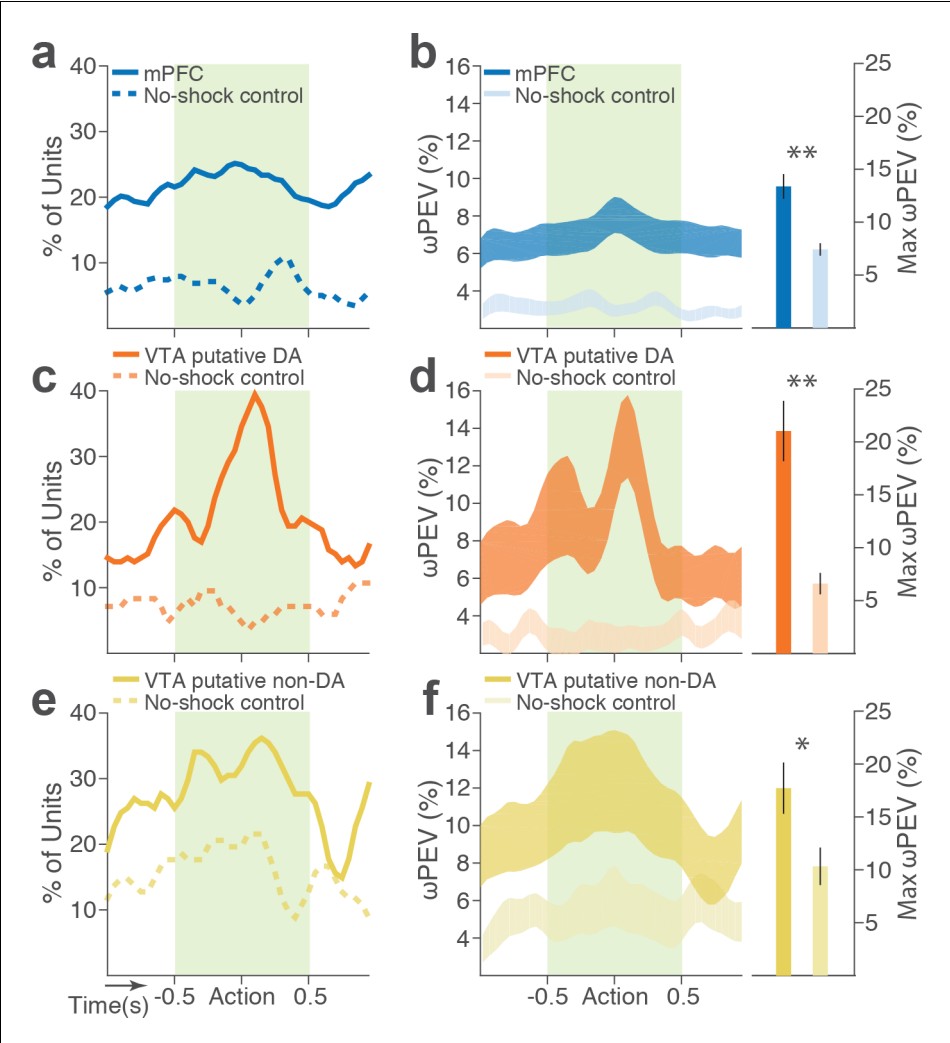

**Figure 5.** Blockwise firing rate changes in the presence vs absence of punishment risk. (a) Proportion of mPFC units showing significant firing-rate changes across blocks during the peri-action epoch in the presence vs absence (no-shock control) of punishment. (b) Left, Percent variance in the mPFC unit firing rates explained by the block shift (ωPEV) in the presence vs absence of punishment (mean ±s.e.m.). Right, Maximum peri-action ωPEV of mPFC units differed in the presence vs absence of punishment (Student's t-test, $t_{291}$ = 3.81, p<0.001). (c) Proportion of VTA putative DA units showing significant firing-rate changes across blocks. (d) Left, ωPEV of VTA putative DA units. Right, Maximum peri-action ωPEV of VTA putative DA units ($t_{81}$ = 4.19, p<0.001). (e) Proportion of VTA putative non-DA units showing significant firing-rate changes across blocks. (f) Left, ωPEV of VTA putative non-DA units. Right, Maximum peri-action ωPEV of VTA putative non-DA units ($t_{74}$ = 2.25, p=0.028). *p<0.05, **p<0.005.
DOI: https://doi.org/10.7554/eLife.30056.013

*Figure 6—figure supplements 1–2*). Significant correlations were observed in all four mPFC and three VTA populations comprising 10 to 25 single units subjected to this analysis (*Figure 6—figure supplements 1–2*). Next we looked to see if the neural population trajectories could track behavioral variation within each block. As our behavioral data indicated more pronounced trial-to-trial variability in RT in the riskiest block (*Figure 1—figure supplement 1*), it was of particular interest whether the increased behavioral variability was correlated with the neural population trajectories in this block. Significant correlations were observed specifically in the riskiest block in two of the four mPFC populations (*Figure 6—figure supplement 1*), indicating that the neural and behavioral correlation emerged as the behavioral variability arose along with risk of punishment. Similarly, significant correlations were detected in two of the three VTA populations in the riskiest block (*Figure 6—figure supplement 2*).

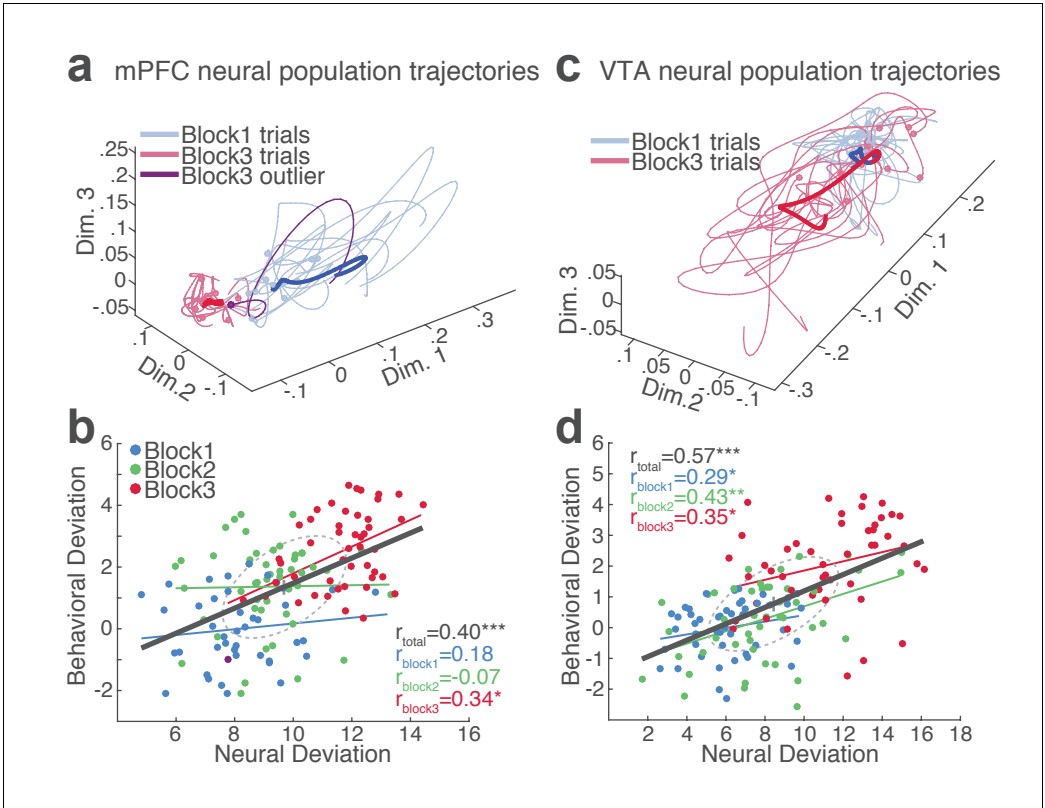

**Figure 6.** mPFC and VTA neuronal populations track the trial-to-trial variability in RT. (**a**) Trial-by-trial trajectories of an mPFC neuronal population (16 units) activity were extracted using the GPFA (Materials and methods). Trajectories of trials in block 1 and 3 are visualized in a population state space comprising the top three ortho-normalized dimensions. For simplicity, ten trials were randomly selected per block. The dimensionality of state space was determined to be five for all populations, based on the cross-validated data likelihood (Materials and methods). Each trajectory corresponds to −0.5 to 0.5 s peri-action epoch (action occurring at time = 0). Filled circles indicate initial points of neural trajectories. Heavy lines indicate the mean trajectory averaged across all trials in each block. The purple-colored block 3 neural trajectory represents a single trial with an outlying RT similar to a block 1 trial. (**b**) A scatter plot indicating behavioral deviation from the block 1 mean RT and neural deviation from the block 1 mean trajectory of the mPFC population shown in (**a**). RT was log transformed for proper scaling. Superimposed color-coded lines indicate regression slopes per block. The dark gray line indicates the regression slope for total trials pooled across blocks. The behavioral and neural correlation coefficients calculated on total trials and trials per block are indicated. *p<0.05, **p<0.005. (**c**) Neural population trajectories of a representative VTA population comprising 10 units. (**d**) A scatter plot indicating behavioral deviation from the block 1 mean RT vs neural deviation from the block 1 mean trajectory of the VTA population shown in (**c**).

DOI: https://doi.org/10.7554/eLife.30056.014

The following figure supplements are available for figure 6:

**Figure supplement 1.** Single-trial analysis of mPFC neuronal population activity reveals the linkage between neural and behavioral variability during risky reward-seeking.
DOI: https://doi.org/10.7554/eLife.30056.015

**Figure supplement 2.** Single-trial analysis of VTA neuronal population activity reveals the linkage between neural and behavioral variability during risky reward-seeking.
DOI: https://doi.org/10.7554/eLife.30056.016

Finally, we computed trial-by-trial trajectories of neuronal population activity recorded from the no-shock control sessions. Consistent with the lack of correlation observed in the absence of risk, there was no significant correlation between mPFC and VTA population activity and RT (*Figure 6— figure supplements 1–2*). These results indicate that distinct neuronal population activity track trial-to-trial variation of reward-seeking behavior in the presence, but not the absence, of punishment risk.

## Punishment risk diminishes neural synchrony between VTA and mPFC

At the neural network level, synchronous oscillations can provide temporal coordination among groups of neurons, which may subserve cognitive and affective functions (*Adhikari et al., 2010*; *Buschman et al., 2012*; *Fries, 2015*; *Karalis et al., 2016a*; *Kim et al., 2012*; *Likhtik et al., 2014*). We, therefore, examined oscillation-mediated neural synchrony during reward-seeking behavior in the absence and presence of punishment risk.

During punishment-free performance in block 1, theta oscillations in the frequency band of 5 to 15 Hz emerged in mPFC and VTA before and during action execution (*Figure 7a–d*). This oscillation was markedly reduced as a function of punishment risk in both regions (*Figure 7e–f*). Of note, this change in theta oscillation observed before and during the action could not be caused by changes in animals' motor activity because they all engaged in the action within this epoch. A significant interaction between punishment and frequency band was detected in LFP power during post-cue and pre-action time periods in both regions (Repeated measures ANOVA, VTA, post-cue, $F_{52, 1092}$ = 5.11; pre-action, $F_{52, 1092}$ = 7.78, $p$ values < 0.001; mPFC, post-cue, $F_{52, 1872}$ = 1.60, p=0.005; pre-action, $F_{52, 1872}$ = 2.29, p<0.001, *Figure 7g–h*). Theta oscillations appeared to be coherent between the two regions during the punishment-free action, but the coherence significantly decreased as a function of punishment (*Figure 7i*). A significant interaction between punishment and frequency band was observed in LFP coherence during the pre-action period (Repeated measures ANOVA, post-cue, $F_{52, 1092}$ = 0.93, p=0.62; pre-action, $F_{52, 1092}$ = 2.45, p<0.001).

To examine mutual influences (directionality) of LFP time series between the two regions, we quantified Granger causal influences (GC) in VTA-to-mPFC and mPFC-to-VTA directions (Materials and methods). During punishment-free action in block 1, the theta oscillation was driven by VTA, as mPFC was GC influenced by VTA significantly greater than the GC influence in the other direction (*Figure 7j*). A significant interaction between directionality and frequency band was observed in GC coefficients in all blocks, indicating that the oscillation directionality varied across frequency bands (*Figure 7j*; Repeated measures ANOVA, Block 1, $F_{25, 700}$ = 2.05, p=0.002; Block 2, $F_{25, 700}$ = 2.38, p<0.001; Block 3, $F_{25, 700}$ = 5.59, p<0.001). Importantly post hoc analysis revealed significantly greater VTA-to-mPFC directionality in frequency bands including the theta band in block 1, and the directionality became less apparent in blocks 2 and 3 (*Figure 7j*, data not shown for block 2). Taken together, these results suggest that the VTA-driven theta oscillation entrains the VTA-mPFC circuit during punishment-free action. Decline in this entrainment may represent punishment risk, since power, coherence, and directionality of the theta oscillation declined as a function of punishment risk. Analyses of no-shock control data revealed that these punishment-dependent changes in the VTA-mPFC theta oscillations do not occur in the absence of punishment (*Figure 7—figure supplement 1*).

## Risk of punishment weakens LFP–spike synchrony within and between PFC and VTA

Synchronous oscillations can provide temporal coordination of spike activity between local and distant neurons and neuronal communication between coherently timed neuron groups (*Fries, 2015*; *Harris and Gordon, 2015*). The presence of such LFP-mediated spike timing coordination was examined by measuring phase-locking of the neuronal spike activity to local and inter-regional theta oscillations.

Within each region, substantial proportion of single units (37% in VTA and 23% in mPFC) showed significant phase-locking to local theta oscillation in the punishment-free block 1. Consistent with the temporally specific increase in the theta spectral power (*Figure 7c–d*), the phase synchrony arose during action from the baseline level in the phase-locked units in both regions (*Figure 8a and d*; Signed-rank test, $p$ values < 0.001). Enhanced phase-locking, albeit to a lesser degree, was observed even in units that failed the Rayleigh's test (*Figure 8a and d*; Signed-rank test, $p$ values < 0.001), indicating widespread influence of the theta oscillation on local neuronal spike timing. The temporal relationship (directionality) between spike outputs and theta oscillation was examined using a time-lagged phase-locking analysis (*Likhtik et al., 2014*; *Spellman et al., 2015*). The spike-LFP phase-locking was recalculated using spike times shifted relative to the local or inter-regional LFP to infer the directionality in the LFP-spike interaction. We found that in block 1 greater proportions of units appeared to be phase-locked with negative time lags. The vast majority of phase-locked units (VTA,

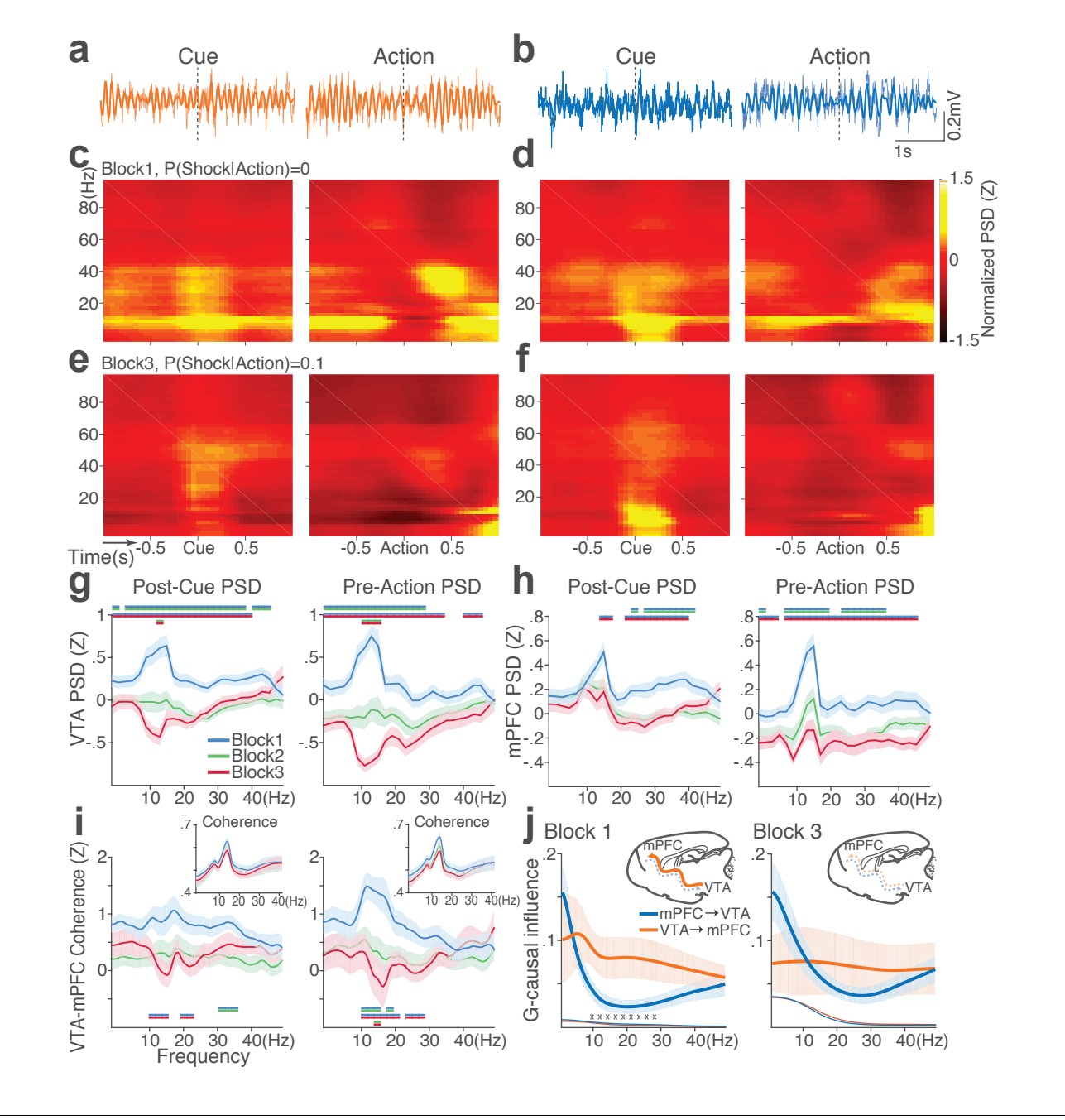

**Figure 7.** Punishment diminishes theta oscillation-mediated neural synchrony in the VTA-mPFC circuit. (a) Representative VTA peri-event LFP traces in a block 1 trial. Bandpass filtered LFP signal (heavy line) is superimposed on the raw trace (thin line). (b) Simultaneously recorded mPFC LFP traces. (c) Baseline-normalized VTA power spectrograms averaged across block 1 trials (left: peri-cue, right: pre-action). mPFC block 1 power spectrograms are in (d). (e) Diminished VTA theta power in block 3. (f) Similar diminishment observed in mPFC theta power. (g) Mean ±s.e.m. (shaded area) normalized VTA PSDs per block corresponding to 1 s post-cue (left) and pre-action (right) epochs. Dual-colored bars indicate significant pairwise differences at corresponding frequency bins according to post hoc analyses (p<0.05). (h) Normalized mPFC PSDs in post-cue (left) and pre-action (right) epochs. (i) Normalized VTA-mPFC LFP coherence in post-cue (left) and pre-action (right) epochs. Insets represent non-normalized LFP coherences of each block. (j) Granger-causality, representing mutual influences (directionality) between VTA and mPFC peri-action LFP time series in block 1 (left) and block 3 (right). Blue and orange curves represent mPFC-to-VTA and VTA-to-mPFC Granger-causal influences, respectively. Shaded areas indicate s.e.m. Thin colored-lines below indicate upper bounds of confidence intervals (α = 0.001) acquired by the random permutation resampling of time bins. Asterisk indicates significant difference between bidirectional Granger-causal influences at the corresponding frequency bin (p<0.05).

DOI: https://doi.org/10.7554/eLife.30056.017

*Figure 7 continued on next page*

*Figure 7 continued*

The following figure supplement is available for figure 7:

**Figure supplement 1.** mPFC and VTA theta oscillations did not change across blocks in the absence of punishment.
DOI: https://doi.org/10.7554/eLife.30056.018

74%; mPFC, 67%) had their maximum phase-locking values (PLVs; Materials and methods) with a negative lag (*Figure 8b and e*; Signed-rank test, *p* values < 0.005). These indicated an entrainment of spike timing by preceding cycles of the oscillation – that is, directionality from the theta oscillation to the spike activity. To examine the modulation of LFP-spike phase-locking by punishment, we compared PLVs across different blocks. A trend toward reduction in PLV was found in block 3 compared with block 1 in mPFC (*Figure 8c*; Signed-rank test, p=0.077), and a significant reduction was found in VTA (*Figure 8f*; p=0.006). Likewise, a trend toward reduction in the proportion of phase-locked

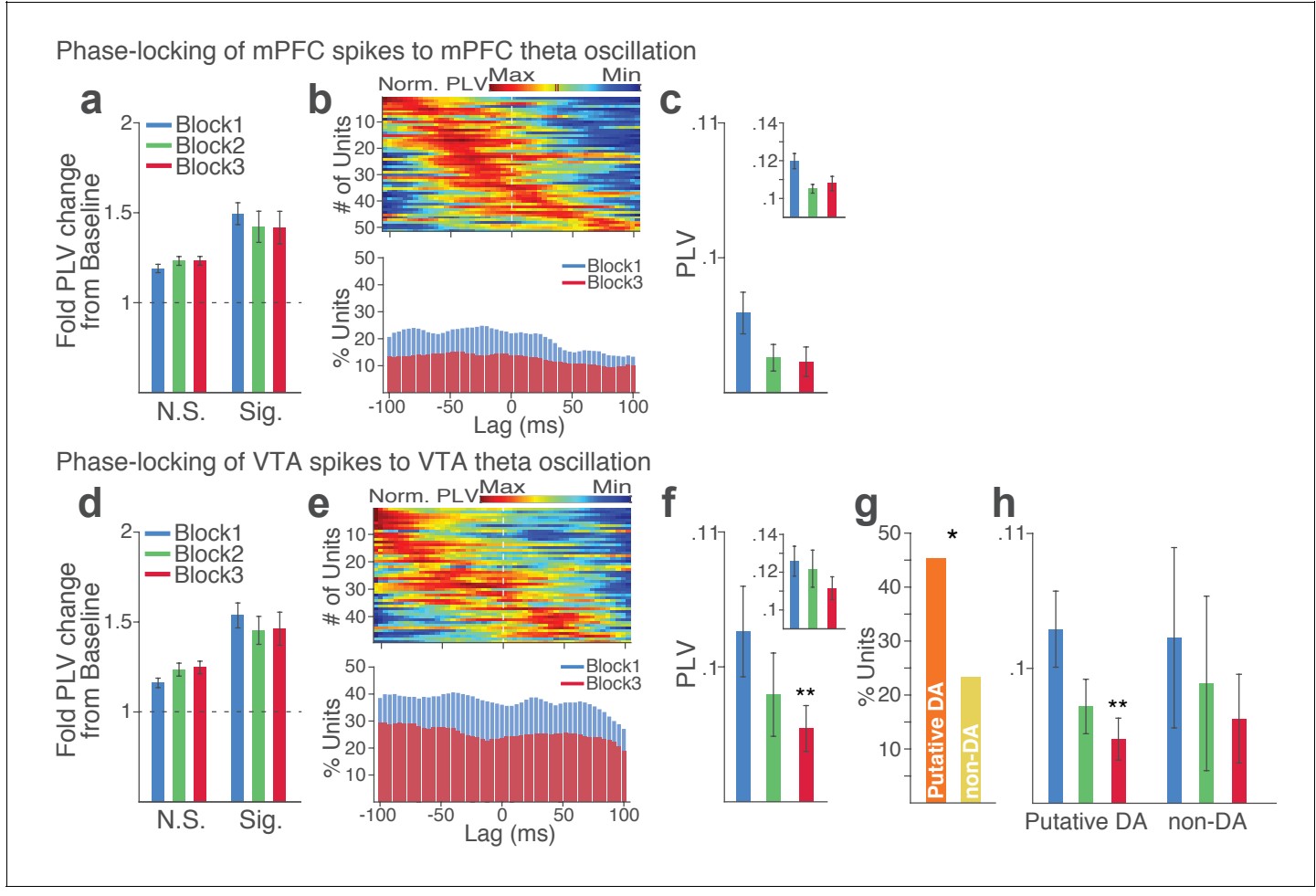

**Figure 8.** Punishment risk reduces VTA and mPFC neuronal synchrony to local theta oscillation. (a-c) Modulation of mPFC neuronal synchrony to mPFC theta oscillation. Phase-locking values (PLVs) were quantified by averaging 1000 mean resultant lengths (MRLs) of the circular phase angle distribution comprising 100 resampled spikes per iteration (Materials and methods). (a) Fold change from baseline in the strength of the neuronal phase-locking during peri-action epoch in units that passed Rayleigh z-test (Sig.) and rest of the units (N.S.). (b) Top, Normalized PLVs in block 1 across a range of time lags for all phase-locked mPFC units, aligned by peak lags. Bottom, Percentage of significantly phase-locked mPFC units in block 1 vs 3 across a range of lags. (c) Mean ±s.e.m. PLVs across different blocks. Inset, PLVs including significantly phase-locked units only. (d-h) Modulation of VTA neuronal synchrony to VTA theta oscillation. (d) Fold change from baseline in the strength of the neuronal phase-locking. (e) Top, Normalized PLVs in block 1 of all phase-locked VTA units. Bottom, Percentage of significantly phase-locked VTA units. (f) Mean ±s.e.m. PLVs across different blocks. (g) Percentage of phase-locked VTA putative DA and non-DA units. (h) PLVs of VTA putative DA and non-DA units plotted separately.
DOI: https://doi.org/10.7554/eLife.30056.019

units was observed in block 3 compared with block 1 in mPFC (*Figure 8b*; Chi-square test, $\chi^2_1$ = 3.25, p=0.071), and a significant reduction was found in VTA (*Figure 8e*; $\chi^2_1$ = 4.31, p=0.038). We next examined VTA putative DA and non-DA neuronal phase-locking separately. Greater fraction of putative DA units (45%) appeared to be phase-locked compared with non-DA units (23%) in block 1 (*Figure 8g*; Chi-square test, $\chi^2_1$ = 5.04, p=0.025). The DA neuronal PLV in block1 significantly declined as a function of punishment contingency in block 2 and 3 (*Figure 8h*; Signed-rank test, *p* values < 0.01), whereas non-DA neuronal PLV did not differ across blocks (*p* values > 0.43). These indicated that the punishment risk-induced reduction in the VTA neuronal phase-locking was predominately due to the reduction in DA neuronal synchrony.

Next we examined LFP-spike phase-locking between VTA and mPFC. Based on the Granger causal influence indicating VTA-to-mPFC directionality in theta oscillations, we anticipated a stronger mPFC neuronal synchrony to VTA than that of the other direction. Consistent with this, we found that a substantial proportion of mPFC units (31%) were phase-locked to VTA theta oscillation in block 1. A representative mPFC unit with significant phase-locking is shown in *Figure 9a–b*. The inter-regional spike-phase synchrony emerged during action compared to baseline (*Figure 9c*, Signed-rank test, *p* values < 0.001). We then examined directionality of the LFP-spike synchrony using the time-lagged phase-locking analysis. In block 1, the majority of phase-locked units had their peak PLVs with a negative lag (*Figure 9d*; Signed-rank test, p=0.066). Likewise, greater proportions of phase-locked units were observed on negative lags (*Figure 9d*, bottom). In addition, the mean PLV across negative time lags appeared to be greater than that of the positive lags (*Figure 9e*; Signed-rank test, p=0.023). These indicate that mPFC neurons are entrained by the preceding VTA theta oscillatory cycles suggesting VTA-to-mPFC directionality. When compared across blocks, the mPFC neuronal entrainment by the VTA theta oscillation declined as a function of punishment contingency (*Figure 9e–g*, Signed-rank test, p=0.003). As the degree of phase-locking diminished, the VTA-to-mPFC directionality also declined (*Figure 9d–e*).

Finally, we examined VTA neuronal phase-locking to the mPFC theta oscillation. The degree of VTA neuronal phase-locking to mPFC theta oscillation was weaker than the mPFC neuronal phase-locking to the VTA theta oscillation (Wilcoxon Rank-sum test, p<0.001), indicating VTA-to-mPFC directionality in the theta oscillation-mediated spike phase modulation (*Figure 9—figure supplement 1*). The PLVs did not differ across different blocks in both putative DA and non-DA units (*Figure 9—figure supplement 1*). Analyses of no-shock control data showed no change in neural synchrony across blocks in the absence of punishment (*Figure 9—figure supplements 2–3*).

Collectively, these analyses indicate that a coherent theta oscillation transiently synchronizes the VTA-mPFC neural network during reward-guided actions and that this synchrony declines when there is a risk of punishment associated with those actions (*Figure 9h*).

## Discussion

Previous work has implicated mPFC and VTA neural networks in representation of actions that are associated with either reward or punishment. In reality, however, an action rarely retains such unitary outcome. In particular, reward-guided actions often involve varying risk of punishment. This risk must be computed by the brain for optimal selection and execution of actions. Here we addressed how mPFC and VTA represent risk of punishment during rewarded actions by simultaneously recording from both regions as animals performed an instrumental task where an action consistently procured a reward but probabilistically led to punishment. We found that these structures use multiple coding schemes, involving spike-rate and LFP-mediated neural synchrony, to compute risk of punishment. At neuronal ensemble level, putative DA and non-DA VTA and mPFC units responded to the same action differently if that action was punishment-free versus punishment-prone, suggesting that these neurons encode the relationship between action and punishment. At the network level, coherent theta oscillations, which synchronized mPFC and VTA activity during punishment-free actions, declined during punishment-prone actions, indicating that risk of punishment disrupts the mPFC-VTA synchrony that emerges during reward seeking behavior.

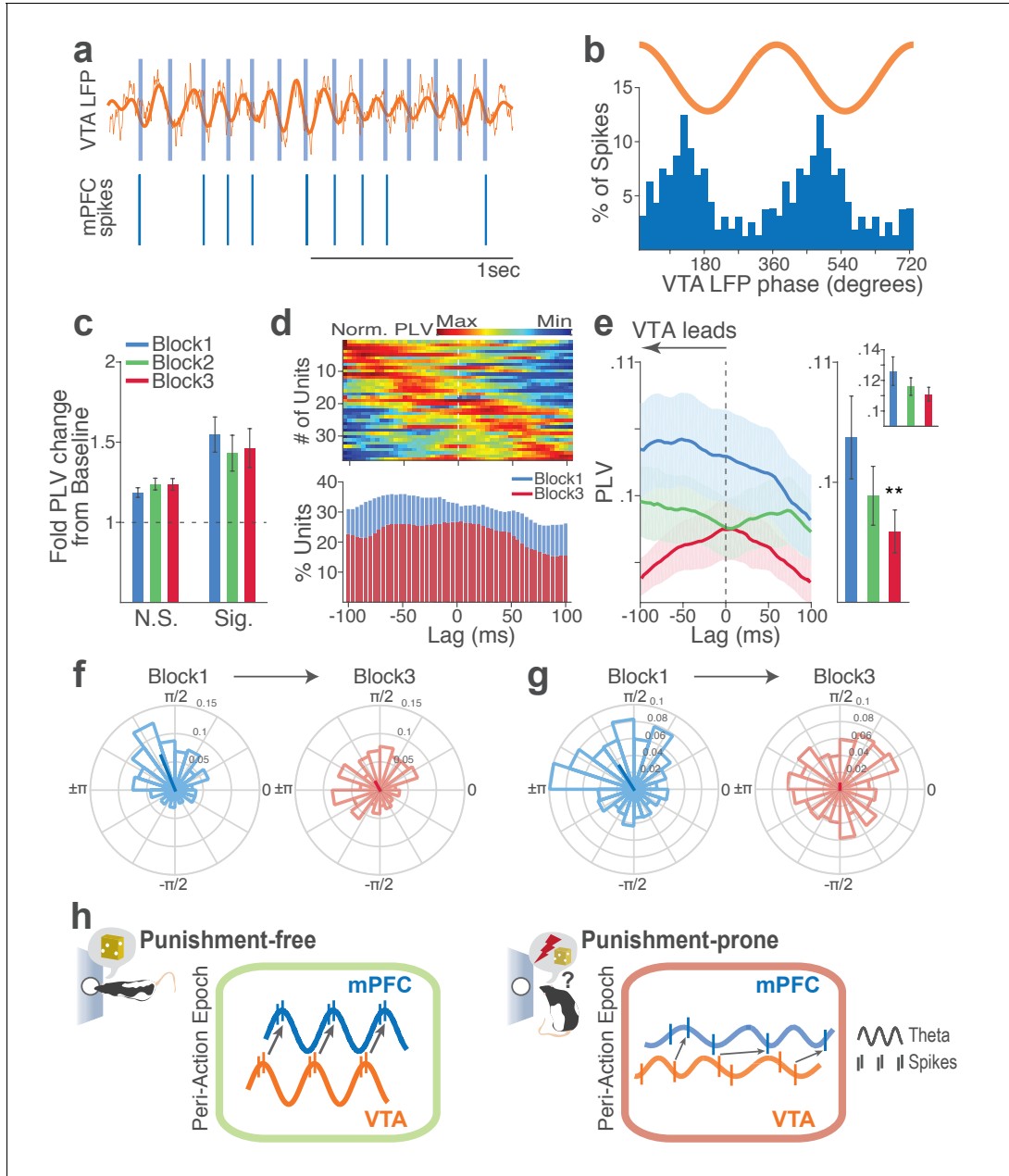

**Figure 9.** Punishment risk reduces mPFC neuronal synchrony to the VTA theta oscillation. (**a**) Top, Example raw (thin line) and bandpass filtered (heavy line) VTA LFP traces. Bottom, Neuronal spikes of a simultaneously recorded mPFC single unit. This unit's preferred phase is indicated with light blue columns superimposed on the LFP trace. (**b**) Distribution of spike phase angles of the example mPFC unit relative to the VTA theta oscillation (Rayleigh's p<0.001). (**c**) Fold change from baseline in the strength of mPFC neuronal phase-locking (PLV) during the peri-action epoch in units that passed Rayleigh z-test (Sig.) and rest of the units (N.S.). (**d**) Top, Normalized PLVs in block 1 across a range of time lags for all phase-locked mPFC units, aligned by peak lags. Bottom, Percentage of significantly phase-locked units in block 1 vs 3 (**e**) Left, PLVs calculated with negative and positive time lags applied to spike trains relative to LFP time series. Right, Mean ±s.e.m. PLVs of all units across different blocks. Inset, PLVs including significantly phase-locked units only. (**f-g**) Each polar plot represents the distribution of spike-phase angles of an example mPFC unit relative to VTA theta oscillation in block 1 vs 3. To quantify the circular concentration of phase angles, we calculated the mean resultant vector indicated as a superimposed bar on each polar plot. (**h**) At the neural circuit level, the theta-oscillation-mediated neural synchrony in the VTA-mPFC circuit that emerged during punishment-free actions declined during punishment-prone actions. Neural synchrony mediated by the theta oscillation may subserve binding of the VTA-mPFC neurons responding to the appetitive action into the 'appetitive' neural network. Our observation of decline in theta-mediated neural synchrony may reflect reduced activation of the appetitive neural network in the presence of punishment risk.

DOI: https://doi.org/10.7554/eLife.30056.020

The following figure supplements are available for figure 9:

*Figure 9 continued on next page*

*Figure 9 continued*

**Figure supplement 1.** VTA single units show weak phase synchrony to the mPFC theta oscillation.

DOI: https://doi.org/10.7554/eLife.30056.021

**Figure supplement 2.** mPFC neuronal synchrony to mPFC and VTA theta oscillations did not change across blocks in the absence of punishment (No-shock control).

DOI: https://doi.org/10.7554/eLife.30056.022

**Figure supplement 3.** VTA neuronal synchrony to VTA and mPFC theta oscillations did not change across blocks in the absence of punishment (No-shock control).

DOI: https://doi.org/10.7554/eLife.30056.023

## VTA and mPFC neurons represent risk of punishment on different timescales

Using a novel behavioral paradigm that assessed punishment risk during reward-guided behavior, we found that the majority of VTA and mPFC neurons respond to punishment risk by modulating their firing rates. Critically, this response was most pronounced at the time of the action as compared to other task events such as cue or reward delivery. This suggests that neurons in both structures preferentially encode risk of punishment during reward-seeking actions.

Given the relatively low probability and modest experience of punishment in the present task, one may wonder to what extent animals were cognizant of the punishment contingency on the action. Our data suggest that the punishment based behavioral changes is due to both learning of action-punishment contingency and state-dependent effects of punishment. Learning of contingency is supported by data showing a selective increase in instrumental RTs but not in reward RTs. As shown in the video reproducing stereotypical risky actions, animals were timid/cautious at the time of instrumental actions and made many 'incomplete' nose pokes, which was not the case for nose pokes to the food trough. The latter is mechanically the same action to which punishment was never contingent on. These suggest that pronounced changes in neural activity within the peri-action epoch may reflect changes in action-punishment contingency. We also observed a correlation between behavior and neuronal responses as behavioral variability increased with risk of punishment suggesting that the direct impact of shock experiences may contribute, in part, to behavioral and neural changes irrespective of learned contingency.

Punishment influences behavior, and neuronal representation associated with that behavior, on multiple timescales (*Cohen et al., 2015*; *Somerville et al., 2013*). On short timescales, real-time neural processing of punishment may be important to signal contingency of punishment on a specific event in order to promote rapid behavioral adaptation. Our data suggest that VTA DA neuronal signaling may be involved in this context. DA neurons displayed phasic excitatory responses tightly linked to each of the task events (cue, action, and reward). Importantly, the DA neuronal encoding of punishment was concentrated around the time of the action compared with other task epochs, suggesting that DA neuronal signaling of punishment may primarily reflect the action-punishment contingency. On longer timescales, punishment can elicit persistent changes in motivational and emotional states – e.g., changes in mood. We found that mPFC and VTA non-DA neurons display temporally diffuse encoding of punishment within the peri-action window. Likewise, many of the mPFC and non-DA neurons showed significant modulation of their baseline firing rates, suggesting that these neurons may encode punishment with persistent changes in activity. This sustained change in activity may be responsible for longer-lasting affective impact of punishment.

## Neurons display bidirectional responses to punishment risk

Subpopulations of VTA putative DA, non-DA, and mPFC neurons responded to punishment risk by increasing or decreasing their peri-action firing rates. This is not surprising given that while punishment is an aversive event that should be avoided, it is highly salient and deserving prioritized attention. These aspects of punishment need to be represented for appropriate behavioral coping. In general, the direction of neuronal responses to appetitive vs aversive events is thought to carry information about motivational properties encoded by the neuronal activity. While heterogeneous response patterns have been widely observed in PFC neurons that respond to punishment (*Kobayashi et al., 2006*; *Matsumoto et al., 2007*; *Seo and Lee, 2009*; *Ye et al., 2016*), there is

some debate on whether the VTA DA neurons respond to punishment with excitatory or inhibitory responses (*Bromberg-Martin et al., 2010*; *Schultz, 2016*). It has been demonstrated that reward prediction error (RPE)-coding DA neurons respond to appetitive (better-than-predicted) vs aversive (worse-than-predicted) events by excitation and inhibition, integrating information about appetitive and aversive events into a common currency of value (*Eshel et al., 2016*; *Matsumoto et al., 2016*; *Mileykovskiy and Morales, 2011*; *Mirenowicz and Schultz, 1996*; *Roitman et al., 2008*). We found that a subset of VTA putative DA neurons conformed to this pattern. These neurons responded to punishment-free (purely appetitive) actions with phasic excitation, which decreased as a function of punishment risk.

In contrast, a greater proportion of putative DA neurons showed excitatory responses to punishment; that is, they treated appetitive and aversive components in the same direction, and responded to actions prone to punishment with further excitation. We observed that excitatory representation of punishment risk was predominant among DA neurons, suggesting that risk of punishment is not simply encoded as reduced value of the action. Previous studies have reported that a subpopulation of putative DA neurons show similar excitatory responses to appetitive and aversive stimuli (*Brischoux et al., 2009*; *Joshua et al., 2008*; *Matsumoto and Hikosaka, 2009*; *Valenti et al., 2011*). An excitatory DA neuronal encoding of aversive or neutral events has been interpreted as conveying motivational salience (*Bromberg-Martin et al., 2010*; *Matsumoto and Hikosaka, 2009*), detection (*Nomoto et al., 2010*), intensity (*Fiorillo et al., 2013*) of a sensory event, as well as generalization effect of rewarding stimuli (*Kobayashi and Schultz, 2014*). Combination of these factors may comprise the excitatory DA neuronal encoding of punishment-prone actions we observed. Furthermore, the DA neuronal encoding of aversion has been suggested to depend on animals' behavioral state. A recent study demonstrated that DA neurons encoded aversive events with inhibition in a low reward context but with biphasic responses in a high reward context (*Matsumoto et al., 2016*). In addition, different DA neural responses have been reported based on how animals responded to an aversive event. For example, DA concentration in the ventral striatum increased or decreased when rats displayed active avoidance or passive reaction (freezing) to shock-predicting cues (*Oleson et al., 2012*; *Wenzel et al., 2015*). These patterns of state dependency in DA neural responses to aversion are in line with our observation of excitatory encoding of punishment when risk was taken for reward in a highly rewarding context. Thus, our results support the view that DA neuronal signaling of aversion is not a uniform process and depends on behavioral states and reward availability in the environment.

## VTA-mPFC neural synchrony declines with punishment

At the network level, we observed that coherent theta oscillation synchronizes VTA and mPFC specifically during punishment-free actions, effectively phase-modulating the neuronal spike activity in the two regions. Analyses of the temporal relationship indicated that the neural synchrony arose in the VTA-to-mPFC direction. That is, VTA-driven theta oscillation entrained mPFC LFPs and neuronal spike activity during punishment-free actions. The theta oscillation preferentially entrained putative DA neurons but much fewer non-DA neurons in the VTA. Considering the phasic excitatory responses of DA neurons during action, the theta oscillation-mediated neural synchrony may promote phase-coupling between VTA DA and mPFC neurons selectively during reward-seeking behavior when there is no risk of punishment. This phase-coupling diminished during punishment-prone actions. Thus, prediction of punishment can be inferred by dual alterations in the phase and the rate of the DA neurotransmission in the mPFC (model in *Figure 9h*). Our data showing preferential DA (vs non-DA) neuronal phase-locking with VTA and mPFC theta oscillations supports the theoretical model suggesting that the VTA DA input may play a crucial role for cortical theta oscillations (*Buhusi and Meck, 2005*). In addition, our data is consistent with previous studies showing that inhibiting DA D1 receptors in mPFC diminishes theta oscillations (*Parker et al., 2014*). Establishing the causality of VTA DA neuron driving mPFC theta oscillation, however, would require future experiments with pharmacological or optogenetic manipulation of specific cell types.

While somewhat out of the scope of the present study, it is noteworthy that theta oscillations in the mPFC and/or VTA have been related to the hippocampal theta oscillation (*Hyman et al., 2005*; *Jones and Wilson, 2005*; *Siapas et al., 2005*; *Sirota et al., 2008*). Local infusion of DA in mPFC enhances theta oscillations and coherence between mPFC and hippocampus (*Benchenane et al., 2010*), while both mPFC and VTA neurons exhibit phase coherence to the hippocampal theta

oscillation (*Fujisawa and Buzsáki, 2011*). Thus, the theta-mediated neural synchrony we observed in the VTA-mPFC circuit may be coupled with the hippocampal theta oscillation, including the possibility that the theta oscillation are led by the hippocampus and/or a third structure.

Recent studies have reported slow (4 Hz) oscillation in mPFC during fear-conditioned freezing in synchrony with other regions including the amygdala (*Dejean et al., 2016*; *Karalis et al., 2016*; *Likhtik et al., 2014*). The distinct behavioral states associated with 4 Hz oscillations in these studies taken together with theta oscillations observed in our study may suggest that mPFC is entrained by different bands of oscillations in appetitive vs aversive states. However, the 4 Hz oscillation in the VTA-mPFC circuit has also been associated with working memory (*Fujisawa and Buzsáki, 2011*). The fast vs slow mPFC oscillations occurring in appetitive vs aversive states may arise in preferential synchrony with VTA or amygdala in appetitive vs aversive states, thereby the bottom-up information transfer from the two subcortical regions can be routed depending on the behavioral context. This scenario would predict that the theta-oscillation-mediated VTA-mPFC synchrony decreases in the presence of punishment while 4 Hz oscillation increases in the mPFC. Consistent with this, we observed that mPFC theta-oscillation-mediated synchrony significantly declined as a function of action-punishment contingency. We did not observe the emergence of 4 Hz oscillation presumably because, unlike fear conditioning paradigms, our task involved instrumental actions. Of note, entrainment of a neural circuit with varying frequency oscillations as a function of a task variable has been widely observed in sensory cortical circuits (*Bosman et al., 2012*; *Jia et al., 2013*). Such frequency modulation, along with the power modulation, could promote selection and binding of task-relevant neuronal ensembles to give rise to a functional neural network (*Fries, 2015*). Likewise, our data may reflect the rise and fall of coherent VTA and mPFC neuronal ensembles that may promote a flexible control of instrumental behavior as a function of punishment contingency.

The neural synchrony mediated by different bands of oscillations in distinct behavioral states may also implicate neuron groups in the other connected structures such as nucleus accumbens, amygdala, hippocampus, and lateral habenula that are critical for appetitive and aversive behaviors. Mounting evidence suggests that distinct neuron groups within these regions selectively respond to appetitive vs aversive events and display different patterns of input-output connectivity (*Howe and Dombeck, 2016*; *Lammel et al., 2012*; *Parker et al., 2016*; *Roeper, 2013*; *Ye et al., 2016*) suggesting that projection specificity may be the foundation for selective recruitment of distinct neuron groups in distinct-valence experiences. The anatomical connectivity alone, however, may not be sufficient to bind the neuron groups tuned to appetitive or aversive events into a functional neural network in a timely manner. Therefore, the neural synchrony mediated by coherent oscillations, such as that demonstrated here, may play a key role for the rise and fall of the functional neural networks depending on the behavioral context.

In conclusion, proper encoding of punishment risk and its contingency on actions and outcomes are fundamental to adaptive behavior and survival. Our data reveal dynamic coding schemes of the VTA-mPFC neural networks in representing risk of punishment and punishment-based modulation of rewarded actions.

## Materials and methods

### Subjects and surgical procedure

Male Long Evans rats weighing 300 ~ 400 g (Harlan) were singly housed on a 12 hr light/dark cycle (lights on at 7 p.m.). All data were collected during the dark cycle. Microelectrode arrays were surgically implanted in ipsilateral mPFC and VTA (N = 10) or bilateral mPFC (N = 4) of isoflurane-anesthetized rats (*Figure 2—figure supplement 1*). All mPFC electrode arrays were placed in the prelimbic subregion of the mPFC. The following coordinates relative to the bregma were used: mPFC = AP +3.0 mm, ML 0.7 mm, DV 4.0 mm; VTA = AP −5.3 mm, ML 0.8 mm, DV 8.2 mm (*Paxinos and Watson, 1998*). Only the mPFC and VTA single units and field potentials simultaneously recorded were subjected to inter-regional network analyses. Behavioral training began after 1 week of postsurgical recovery. At the completion of all recordings, rats were anesthetized with 400 mg/kg chloral hydrate and perfused with saline and 10% buffered formalin. Coronal brain slices of mPFC and VTA were collected and cresyl-violet stained. Placements of electrode arrays were verified under a light microscope. All procedures were in accordance with the National Institute of

Health's Guide to the Care and Use of Laboratory Animals, and were approved by the University of Pittsburgh Institutional Animal Care and Use Committee.

## An instrumental task with varying punishment risk

After the postsurgical recovery, rats were kept at 85% of their free-feeding weight on a restricted diet of 13 g food pellets a day with free access to water. In an operant chamber (Coulbourn Instruments), rats were fully trained to make an instrumental nose poke to the cue port to receive a sugar pellet at the food trough located in the opposite side of the chamber on the fixed ratio schedule of one – that is, FR1 (*Figure 1a–b*). The operant chamber was equipped with an infrared activity monitor (Coulbourn Instruments) which detected animals' movement episodes. Time elapsed without any movements greater or equal to 1 s was scored as immobility and summed to compute the immobile RT (*Figure 1*). After completion of three FR1 sessions consisting of 150 trials in 60 mins, rats underwent a no-shock control recording session, in which they performed 150 trials of nose pokes divided into three blocks in the absence of punishment risk (*Figure 1d*). Then rats were trained with the task consisting of three blocks with varying degrees of action-punishment contingency (50 trials per block). Each block was assigned an action-punishment contingency of 0, 0.06, or 0.1 – that is, the conditional probability of receiving an electrical foot shock (0.3 mA, 300 ms) given an action. The action–reward contingency was kept at one across all training and recording sessions; that is, every nose poke procured a reward even in the shock trials. To minimize generalization of the action-punishment contingency across blocks, they were organized in an ascending shock probability order – Block1: 0, Block2: 0.06, Block3: 0.1, interleaved with 2 min timeout between blocks. In block 2 and 3 of each session, 3 and 5 trials were pseudo-randomly selected and followed by an electrical foot shock. No explicit cue was provided on shock trials to keep the shock occurrence unpredictable. The cue onset only signaled initiation of a trial. Animals were informed of the block shift by the 2 min darkened timeout in between blocks. In addition, the first shock trial of block 2 and the first two shock trials of block 3 were randomly selected from the initial 5 trials of each block. Also, animals completed two sessions of this task before the recording session, thus the shock occurrence and the task design including the ascending punishment contingency were not novel to them at the time of the recording session. All training and recording sessions were terminated if not completed in 180 mins, and data from the completed sessions only were analyzed. Animals displayed stable behavioral performance overall without any sign of contextual fear conditioning, since they performed fearless in the safe block across all sessions. In addition, there was no evidence for habituation to the shock as they showed equivalent punishment-based behavioral changes across sessions. For the diazepam pretreatment experiment, a separate group of rats (N = 9) were trained using abovementioned procedure, and they underwent three test sessions with intraperitoneal pretreatment of saline – diazepam (2 mg/kg, Hospira, Inc.) – saline. After injection, animals were returned to their home cage for 10 min before they were placed in the operant chamber. Three days of washout period was allowed between sessions.

## Electrophysiology

Single-unit activity and local field potentials (LFPs) were recorded simultaneously using a pair of eight channel Teflon-insulated stainless steel 50 μm microwire arrays (NB Laboratories). Unity-gain junction field effect transistor headstages were attached to a headstage cable and a motorized commutator nonrestrictive to the animals' movement. Signals were amplified via a multichannel amplifier (Plexon). Spikes were bandpass filtered between 220 Hz and 6 kHz, amplified ×500, and digitized at 40 kHz. Single-unit activity was then digitally high-pass filtered at 300 Hz and LFP were low-pass filtered at 125 Hz. Continuous single-unit and LFP signals were stored for offline analysis. Single units were sorted using the Offline Sorter software (Plexon). Only the single-units with a stable waveform throughout the recording session were further analyzed. If a unit presented a peak of activity at the time of the reference unit's firing in the cross-correlogram, only either of the two was further analyzed.

## Neural data analysis

Single unit and LFP data analyses were conducted with Matlab (MathWorks) and SPSS statistical software (IBM). For single unit data analyses, 1 ms binned spike count matrix of the peri-cue, action, and

reward periods (starting 2 s before each event and ending 2 s after each event) were produced per unit. The baseline period was a 2 s time window beginning 2.5 s before the cue onset. For all neural data analyses, the trials with shock delivery (three and five trials for block 2 and 3, respectively) were excluded as single-unit and LFP signals in these trials were affected by electrical artifacts during shock delivery.

## Trial-averaged firing-rate analysis

Spike count matrices were further binned using a 50 ms rectangular moving window with steps of 50 ms within the −2 to 2 s epoch aligned to the task event occurring at time = 0, and smoothed across five 50 ms time bins for the firing rate analysis. Binned spike counts were transformed to firing rates and averaged across trials. The trial-averaged firing rate of each unit was Z-score normalized using the mean and standard deviation of its baseline firing rate.

## VTA cell classification

The VTA single units were classified into putative dopamine (DA) or non-dopamine (non-DA) neurons based on two criteria. First, units whose mean baseline firing rate slower than 12 Hz, waveform width greater than 1.2 ms were considered as potential DA units (*Grace and Bunney, 1984*; *Kim et al., 2016*; *Schultz and Romo, 1987*). This traditional classification, however, has been suggested to be potentially inaccurate (*Margolis et al., 2006*). Thus, the second criterion utilized the neuronal reward response properties for the putative DA and non-DA cell identification. Receiver-operating characteristic (ROC) curves were calculated by comparing the distribution of firing rates across trials in 100 ms bins (starting 0.5 s before reward delivery and ending 1 s after reward delivery) to the distribution of baseline firing rates. Principal component analysis was conducted using the singular value decomposition of the area under the ROC (auROC). Units were mapped in the 3-d space comprising the top three principal components. Within the 3-d PC space, unsupervised clustering was conducted by fitting Gaussian mixture models using the expectation-maximization algorithm. This method found two clusters: one with phasic excitation to reward (Type 1), one with sustained excitation or suppression to reward (Type 2) (*Figure 2—figure supplement 2c–e*). Units in the former class were classified as putative DA units, as previous studies have shown that optogenetically tagged dopamine neurons displayed similar phasic excitatory reward responses (*Cohen et al., 2012*; *Eshel et al., 2015*). Taken together, we defined a VTA unit satisfying both criteria as a putative DA unit and a unit that met either or none of the criteria as a putative non-DA unit (*Figure 2—figure supplement 2*). mPFC units were not classified based on their firing and spike-waveform properties. Only 2 out of the total 293 mPFC single units had the mean baseline firing rates higher than 20 Hz, thus few fast-spiking interneurons should be included in our data analysis.

## Spike rate selectivity

To quantify single neuronal encoding of blockwise action-punishment contingency, we computed a bias-corrected percent explained variance (ωPEV) statistic with binned spike counts calculated in a 200 ms rectangular window moving with steps of 50 ms within the 2 s peri-event epochs (−1 to 1 s with an event occurring at time = 0).

$$\omega\text{PEV} = \frac{SS_{Blocks} - df_{Blocks}MS_{Error}}{SS_{Total} + MS_{Error}}$$

where $SS_{Blocks}$ and $SS_{Total}$ are the between-blocks (action-punishment contingency) and total sums of squares, $df_{Blocks}$ is degrees of freedom, and $MS_{Error}$ is the mean squared error. This formulation resulted in an unbiased metric with an expected value of zero when there is no difference across blocks (*Buschman et al., 2012*; *Keren and Lewis, 1979*). A unit was determined to encode action-punishment contingency if its peri-event ωPEV surpassed 'the global ωPEV band', which was defined as the upper bound of the 99% confidence interval of the trial-shuffled (1000 times) surrogate ωPEV distribution – that is, fewer than 1% of the trial-shuffled ωPEVs crossed the global band across all time bins in the peri-event epoch (α = 0.01). To find the global ωPEV band, we computed the mean and standard deviation of the trial-shuffled ωPEV distribution. By stepping up from the mean by one-hundredth of the standard deviation, we found the pointwise band at each time bin and the global band across time bins both at α = 0.01 (*Figure 2d–e*). This approach effectively resolves the

issue of multiple comparisons that can arise as statistical comparisons made separately across multiple time bins increase the rate of false rejection of the null hypothesis (*Fujisawa et al., 2008*). We repeated this analysis using the mutual information metric, and found that the two metrics yielded similar results.

## Linear regression analysis

For a standardized quantification of the individual neuronal encoding of action-punishment contingency in peri-event epochs, we computed the standardized regression coefficient of the following linear regression model for each unit:

$$\mathrm{SC} = \beta_{\textit{punishment risk}} \mathrm{x} + \varepsilon$$

where $\mathrm{SC}$ denotes binned spike counts calculated in a 200 ms moving window with steps of 50 ms, $\beta_{\textit{punishment risk}}$ regression coefficients for the independent variable, blockwise punishment contingency (1, shock prob. = 0; 2, shock prob. = 0.06; 3, shock prob. = 0.1), respectively. The regression coefficients were standardized by $\beta \times (\mathrm{S}_x / \mathrm{S}_y)$, where $\mathrm{S}_x$, $\mathrm{S}_y$ denote the standard deviations of independent and dependent variables, respectively.

## Gaussian-process factor analysis (GPFA)

GPFA extracts a smooth low-dimensional neural trajectory from simultaneously recorded neuronal time series data (binned spike counts). GPFA performs smoothing and dimensionality reduction in a common probabilistic framework. The GPFA model simply consists of a set of factor analyzers, one at each time bin, that are linked together in the low-dimensional state space by a Gaussian process. We provide a simple mathematical description of GPFA below. *Yu et al. (2009)* provides a thorough analytical and practical discussion of the GPFA model. Let $y_{:,t} \in \mathbb{R}^{q \times 1}$ be the high-dimensional vector of square-rooted spike counts recorded at time point $t = 1, \ldots, T$, where $q$ is the number of simultaneously recorded single units. We extract a corresponding low-dimensional latent *neural state* $x_{:,t} \in \mathbb{R}^{p \times 1}$, at each time point, where $p$ is the dimensionality of the neural state space (p<q). We define a linear-Gaussian relationship between $y_{:,t}$ and $x_{:,t}$.

$$y_{:,t} | x_{:,t} \sim \mathcal{N}(Cx_{:,t} + d, \, R) \tag{1}$$

where $C \in \mathbb{R}^{q \times p}$, $d \in \mathbb{R}^{q \times 1}$, and $R \in \mathbb{R}^{q \times q}$ are model parameters to be fitted. The neural states $x_{:,t}$ at different time points are linked through Gaussian processes. A separate GP is defined for each dimension of the neural state space indexed by $i = 1, \ldots, p$

$$x_{i,:} \sim \mathcal{N}(0, \, K_i) \tag{2}$$

where $x \in \mathbb{R}^{1 \times T}$ is the $i^{th}$ row of $x_{:,t=1,\ldots,T}$ and $K_i \in \mathbb{R}^{T \times T}$ is the covariance matrix for $i^{th}$ GP. The parameters of the GPFA model (1, 2) were fitted using the expectation-maximization algorithm, which finds the model parameters that maximize the probability of the observed data. Using the learned GPFA model, we extract neural trajectories $\mathrm{E}\left[x_{:,:} | y_{:,:}\right]$ from the observed data. These low-dimensional neural trajectories can be related to the high-dimensional observed activity using (1), which defines a linear mapping $C$ between the two spaces. Each column of $C$ defines an axis in the high-dimensional space, and the $i^{th}$ element in $x_{:,t}$ specifies the location along each axis. Columns of $C$ were orthonormalized for a 'PCA-like' visualization, and the neural trajectories were plotted in the orthonormalized low-dimensional space. The dimensionality was determined based on the distribution of cross-validated data likelihoods across different dimensionalities. We found that data likelihoods peaked or plateaued around the dimensionality of five in all neural populations, thus 5D was used across all populations for better comparability. We also tested other dimensionalities, and found that similar behavioral and neural correlations held at different dimensionalities. GPFA model fitting and extraction of population neural trajectories were implemented using the open-source GPFA matlab code package (*Yu et al., 2009*).

## LFP power spectra and coherence

The local field potential (LFP) power spectral densities were quantified using the chronux routine mtspecgramc (**Bokil et al., 2010**). Briefly, the LFP time series within the peri-event epochs were Fourier transformed in a 500 ms moving window with steps of 50 ms with the multi-taper method applied:

$$S = \frac{1}{K} \sum_{k=1}^{K} \left| \int_0^T u_k(t) x(t) e^{-2\pi i f t} \, dt \right|^2$$

where $u_k(t)$ is the multi-tapers (nine tapers were used), $x(t)$ is the LFP time series in a moving window. The baseline normalized power spectra (Z-score) were calculated using the mean and standard deviation of the baseline power spectra across trials. In addition, we inspected the trial-by-trial spectro-temporal representations of LFP time series applying the continuous wavelet transform. We confirmed that comparable representations were attained by the Fourier- and wavelet-based time-frequency analyses.

The magnitude squared coherence (MSC) between time series recorded from mPFC and VTA was calculated in the same moving window with the nine multi-tapers applied using the chronux routine cohgramc. Briefly, the MSC was quantified as:

$$C_{xy}(f) = \frac{|S_{xy}(f)|^2}{S_x(f) S_y(f)}$$

where $S_{xy}(f)$ is the cross spectral density of LFP time series in the two regions, and $S_x(f), \; S_y(f)$ are the autospectral density for each region.

## Bivariate Granger causality analysis

To examine mutual influences (directionality) between LFP oscillations in the two regions, we quantified Granger causality between the simultaneously recorded peri-action LFP traces (−2 to 2 s around the action occurring at time = 0). The bivariate Granger causality (G-causality) infers causality between two time series data based on temporal precedence and predictability (**Barnett and Seth, 2014**; **Granger, 1969**). That is, a variable $X_1$ 'Granger causes' a variable $X_2$ if information in the past of $X_1$ helps predict the future of $X_2$ with better accuracy than is possible when considering only information already in the past of $X_2$ itself. In this framework, two time series $X_1(t)$ and $X_2(t)$ recorded from mPFC and VTA can be described by a bivariate autoregressive model:

$$X_1(t) = \sum_{j=1}^{p} A_{11,j} X_1(t-j) + \sum_{j=1}^{p} A_{12,j} X_2(t-j) + \varepsilon_1(t)$$

$$X_2(t) = \sum_{j=1}^{p} A_{21,j} X_1(t-j) + \sum_{j=1}^{p} A_{22,j} X_2(t-j) + \varepsilon_2(t)$$

where $p$ is the model order (the maximum number of time-lagged observations included in the model), which was estimated by the Akaike information criterion. We then estimated parameters of the model; $A$ contains the coefficients of the model, and $\varepsilon_1$, $\varepsilon_2$ are residuals (prediction errors) with covariance matrix $\Sigma$ for each time series.

Once the model coefficients $A_j$ and $\Sigma$ are estimated, the spectral matrix can be obtained as:

$$S(f) = \langle X(f) X*(f) \rangle = H(f) \Sigma H*(f)$$

where the asterisk denotes matrix transposition and complex conjugation, $\Sigma$ is the noise covariance matrix, and the transfer function $H(f)$ is defined as the inverse matrix of the Fourier transform of the regression coefficients:

$$H(f) = \left( I - \sum_{j=1}^{p} A_j e^{-2\pi ikf} \right)^{-1} \qquad 0 \leq f \leq 2\pi$$

The spectral G-causality from 1 to 2 is then obtained by:

$$I_{1 \to 2}(f) = -\ln \left( 1 - \frac{(\Sigma_{11} - \Sigma_{21}^2/\Sigma_{22}))|H_{21}(f)|^2}{S_{22}(f)} \right)$$

The spectral G-causality measure lacks known statistical distribution, thus a random permutation method was used to generate a surrogate distribution, by which the upper bound of the confidence interval was found at $\alpha = 0.001$. This procedure was implemented using an open-source matlab toolbox, the Multivariate Granger Causality Toolbox (*Barnett and Seth, 2014*).

## LFP-Spike phase-locking analysis

To quantify the individual neuronal spike time synchrony with the local and inter-regional theta oscillations were quantified as follows. The mPFC and VTA LFPs during the baseline and peri-action 4 s time windows were bandpass filtered to isolate oscillations within 5–15 Hz frequency range. The instantaneous phase of each filtered LFP segment was determined using the Hilbert transform and each spike was assigned the phase of its contemporaneous LFP segment. The phase-locking value (PLV) of each unit was defined as the circular concentration of the resulting phase angle distribution, which was quantified as the mean resultant length (MRL) of the phase angles. The MRL is the modulus of the sum of unit vectors indicating instantaneous phases of each spike occurrence normalized by the number of spikes, thus the MRLs were bound to take a value between 0 (no phase-locking) to 1 (perfect phase-locking).

$$MRL = \frac{1}{N} \left| \sum_{n=1}^{N} e^{j\Phi_n} \right|$$

where $\Phi_n$ represents the phase assigned to $n^{th}$ spike occurrence, and $N$ is the total number of spikes. Since the MRL statistic is sensitive to the number of spikes, we calculated MRL 1,000 times with 100 spikes of each unit randomly selected for each iteration, and the PLV was the MRL averaged across all iterations. As comparing PLVs across blocks with varying action-punishment contingency was of our central interest, PLV was computed per each block. Only the units with their peri-action spike counts within each block greater than 100 in all three blocks were included in the phase-locking analysis. Units passing the Rayleigh's z-test at $\alpha = 0.05$ were determined to be significantly phase-locked. The directionality of the LFP and spike phase relationship was inferred by a time-lagged phase-locking analysis, in which the spike times were shifted relative to the LFP time series, stepping by 4 ms within the range of −100 to 100 ms (*Likhtik et al., 2014*; *Spellman et al., 2015*). At each time lag, the PLV of each single unit and its significance were assessed, and the maximum PLV across all time lags was found for each unit. We repeated the analysis with different time lags and analysis windows, and confirmed that the results were very similar across different parameters.

## Statistical analysis

Parametric statistical tests were used for z-score normalized data and non-normalized data that are conventionally tested using a parametric test. Nonparametric approaches, such as conventional nonparametric tests or bootstrapping were used for a hypothesis test of data, of which statistical distribution is unknown, e.g. phase-locking values (PLVs). For all tests, the Greenhouse–Geisser correction was applied as necessary due to violations of sphericity. All statistical tests were specified as two-sided. Multiple testing correction was applied for all tests including multiple comparisons using the Bonferroni correction.

## Acknowledgement

We thank Byron Yu, Sara Morrison, and members of BM laboratory for critical discussion and reading of this manuscript. We also thank Jesse Wood and Nicholas Simon for technical support for designing the task, and Alberto Del Arco for surgical assistance.

## Additional information

### Funding

| Funder | Grant reference number | Author |
|---|---|---|
| National Institute of Mental Health | R56MH084906 | Bita Moghaddam |

The funders had no role in study design, data collection and interpretation, or the decision to submit the work for publication.

### Author contributions

Junchol Park, Conceptualization, Data curation, Software, Formal analysis, Validation, Investigation, Visualization, Methodology, Writing—original draft, Writing—review and editing; Bita Moghaddam, Conceptualization, Resources, Supervision, Funding acquisition, Visualization, Methodology, Project administration, Writing—review and editing

### Author ORCIDs

Junchol Park [iD] http://orcid.org/0000-0002-4739-0793
Bita Moghaddam [iD] http://orcid.org/0000-0002-5205-417X

### Ethics

Animal experimentation: All surgical and experimental procedures were in strict accordance with the National Institute of Health's Guide to the Care and Use of Laboratory Animals, and were approved by the University of Pittsburgh Institutional Animal Care and Use Committee (Protocol #: 15065884).

### Decision letter and Author response

Decision letter https://doi.org/10.7554/eLife.30056.025
Author response https://doi.org/10.7554/eLife.30056.026

## Additional files

### Supplementary files

• Transparent reporting form
DOI: https://doi.org/10.7554/eLife.30056.024

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
