## [Decision Letter]

Thank you for submitting your article "Discrete and coordinated encoding of punishment contingent on rewarded actions by prefrontal cortex and VTA" for consideration by *eLife*. Your article has been favorably evaluated by Sabine Kastner (Senior Editor) and three reviewers, one of whom, Geoffrey Schoenbaum (Reviewer #1), is a member of our Board of Reviewing Editors. The following individual involved in review of your submission has agreed to reveal their identity: Kay M Tye (Reviewer #3).

The reviewers have discussed the reviews with one another and the Reviewing Editor has drafted this decision to help you prepare a revised submission.

Summary:

In this study, the authors record from MPFC and VTA in rats nosepoking for a food reward on an FR1 schedule in the face of possible punishment. Recordings are made across three 50-trial blocks. In block 1, there is no punishment. In block 2, there is one shock delivered in the initial 5 trials and two more delivered in the next 45 trials. In block 3, there are 2 shocks delivered in the first 5 trials and 3 more delivered in the next 45 trials. Two minute intervals are inserted between blocks to cue the rats, and the rats received 2 training sessions prior to recording. Rats exhibited longer latencies to nosepoke in the punishment blocks, which disappeared if no shock was given (Figure 1) or with diazepam pretreatment. Reward retrieval latencies were unaffected. In this context, they report that neural activity in both regions is modulated by shock frequency (block), particularly around the time of action. Putative dopamine neurons showed this effect strongly, while also responding to the trial cue and reward. Finally oscillations in both areas showed reduced coherence in punishment blocks. The authors conclude that both areas are important for incorporating possible punishment into a reward-seeking behavior, and that oscillatory drive from VTA to MPFC may be important for their coordination.

Essential revisions:

The reviewers were in agreement that there is a major question over whether the rats understand the contingency between nose poking and punishment given the low probability and modest experience they have or whether the effects of punishment are more of a state dependent effect. In other words, would the same effect have been achieved by delivering increasing numbers of non-contingent shocks in the blocks? The reviews mention several ways to address this analytically in the current data or by running a behavioral group and/or recording group with non-contingent shock. If it cannot be clearly established that the current results reflect contingency and/or the authors do not want to run more subjects, then it might be acceptable to refrain from making strong claims about this in the Introduction and Results, and then discuss the possibilities in the Discussion. The current design is obviously appropriate to model predator risks during foraging anyway, as predators are not contingent on picking a berry necessarily. But in this case, the reviewers want to emphasize that the changes to the way this is described in the Introduction and Results must be substantive, removing up front claims that there is an action-punishment contingency. It was also felt that a finer grained analysis of individual rats' behavior and neural activity might be useful in addressing this question and more generally. Other important, but less essential points are outlined below.

Title: According to *ELIFE* guidelines, I believe it needs to mention the model species.

*Reviewer #1:*

In this study, the authors record from MPFC and VTA in rats nosepoking for a food reward on an FR1 schedule in the face of possible punishment. Recordings are made across three 50-trial blocks. In block 1, there is no punishment. In block 2, there is one shock delivered in the initial 5 trials and two more delivered in the next 45 trials. In block 3, there are 2 shocks delivered in the first 5 trials and 3 more delivered in the next 45 trials. Two minute intervals are inserted between blocks to cue the rats, and the rats received 2 training sessions prior to recording. Rats exhibited longer latencies to nosepoke in the punishment blocks, which disappeared if no shock was given (Figure 1) or with diazepam pretreatment. Reward retrieval latencies were unaffected. In this context, they report that neural activity in both regions is modulated by shock frequency (block), particularly around the time of action. Putative dopamine neurons showed this effect strongly, while also responding to the trial cue and reward. Finally oscillations in both areas showed reduced coherence in punishment blocks. The authors conclude that both areas are important for incorporating possible punishment into a reward-seeking behavior, and that oscillatory drive from VTA to MPFC may be important for their coordination.

Overall the study was interesting and presented a number of new results that potentially fit with and extend the current literature. These include the finding that reward responsive dopamine neurons are modulated by punishment. As the authors note in their discussion, this can be accounted for in a variety of ways, but the ongoing dispute over this question makes this result alone particularly important to me. It is probably outside the scope of what they are interested in and it may not be possible in the context of their task, but it would be obviously interesting to know if the variance in this response reflects punishment per se or changes in punishment, and how it relates to RPEs (versus just reward responses). It is also very interesting to see the changes in oscillations and potential impact on single unit signaling.

For me the overriding question is whether the rats understand the contingency between their behavior and the punishment or whether the behavior and resultant neural changes reflect a change in anxiety state independent of contingency. This is an issue, because the probability of punishment is extremely low given the number of trials and amount of training. This is not inappropriate, since the authors predicate the significance of their model on foraging and the example that reward seeking often involves risk. In this context, I think it is not critical that the risk be contingent on the action. It is enough that it is contingent on the rats' presence in the block or the environment. So one way to address this is to change the language in the paper to reflect this distinction, deal with this issue forthrightly in the Discussion, etc. Indeed it is a bit unclear whether the authors really mean to make the strong argument one way or the other.

Alternatively, the authors could do more to show behaviorally and also neurally that the findings reflect a learned action-outcome contingency. This is difficult to show definitively, but you might find clues by examining behavior during the pre-recording sessions to show that the punishment-associated changes increase with learning and become more closely tied to the actions. It might also be of interest to look at behavior and neural activity immediately after a punished trial (i.e. trial N+1, N+2, N+3, versus trials far removed but in the same block). Changes that reflect a true knowledge of the random 6% or 10% contingency should be stable. And obviously, adding recordings made in sessions with non-contingent punishment would also be amazingly interesting. This comparison would provide a potential additional level of detail at which the question could be asked, one that I think would increase the elegance of the work substantially no matter the outcome, but particularly if such sessions showed behavioral differences and/or neural distinctions. For example, what if the current behavior develops with learning, but neural activity recorded in the true non-contingent sessions was exactly as reported here? This would show in fact that the modulations of the dopamine activity occurred in the absence of contingency. I think this would alter the interpretation of these changes.

*Reviewer #2:*

The manuscript 'Discrete and coordinated encoding of punishment contingent on reward actions by prefrontal cortex and VTA' by Park and Moghaddam sought to examine spike and LFP activity in the mPFC and VTA during a reward guided action task that contains three blocks of 50 trials each with 0, 0.06 and 0.1 action-punishment probabilities. The data reported in the manuscript show that RT to initiate an action increases as does immobility with increase probability of punishment across training blocks. Single units in the mPFC and VTA encoded the increase in punishment probability at time of action. Theta oscillations were found to be synchronized between the mPFC and VTA in a VTA>>mPFC direction during the rewarding 0 punishment trials, but this synchronization was disrupted when punishment was added. Further analyses showed phase modulation of neuronal spike activity in the mPFC and VTA in reward but not punishment sessions. The manuscript presents nice evidence for VTA>>mPFC directionality in theta oscillations using Granger causal inferences and phase-locking analyses.

The data are interesting and add to a growing body of research implicating VTA DA neurons in the integration of appetitive and aversive information. The manuscript goes further by linking the VTA and mPFC during reward action and showing that this link is disrupted when punishment is added on the background of reward. The manuscript is certainly strong and the neural analyses impressive. Below I have outlined some concerns/questions:

What is the question the manuscript is asking and attempting to answer? This isn't clear.

A substantial problem lies in the experimental design and the reference to the punishment blocks as action-punishment contingency. The wording suggests that the manuscript is explicitly examining an action followed by an aversive event. However, there is no evidence that the behaviour and neural changes seen across blocks are due to a 'contingency'. Given the few punishment trials (3/50 in block 2 and 5/50 in block 3), it is possible that random shock delivery could yield the same behavioural and neural data. Unfortunately, in the absence of these data, the conclusions would need to be altered. I feel it would be better to stick to more appropriate wording that describes the changes in the blocks as increased punishment probability on the background of rewarded actions with no claims whether the punishment needs to be contingent on the action. As a side note, the VTA DA signal around the action may represent an altered reward value as a result of shock delivery within a block (which is different to representing an action-punishment contingency).

What was the temporal firing of VTA non-DA neurons relative to VTA DA neurons? Is there an interaction between those two types of VTA neurons in the present context?

It is hard to know if the rats knew about the upcoming shock probabilities in Block 2 and 3 or if they learned about those on the fly (i.e. state vs. learning effects). Data from Block 2 would be particularly telling in this regard and I think more data from that block should be added to clarify the question. For example, behavioural and neural data from the start of block 2 prior to first shock delivery would provide evidence whether the rats entered block 2 knowing about the increased shock probability (as argued in the manuscript) or behaved similarly to block 1 and altered their behaviour following the first shock presentation. With regard to the third block, the delivery of shock in the second block would likely have been a predictor for the upcoming increase in shock probability. This too can be inferred from the start of block 3 prior to first shock delivery in the form of increased RT.

*Reviewer #3:*

Overall, I am enthusiastic about this study – though I do have some suggestions for improving the manuscript. This work is of interest, adding to the growing literature of the role of VTA-DA neurons and its relationship to the mPFC and aversively-motivated behavior. It certainly has many merits including the novel behavioral paradigm and the simultaneous recordings in PFC and VTA. Perhaps the most intriguing finding, is that punishment contingencies increase the number of putative VTA-DA units that are excited at the time of action (nose-poke). It is unfortunate that the manuscript cannot shed more light on the nature of the precise connections between the VTA and mPFC or the causal relationship between their findings and behavioral output. That said, I believe this manuscript could be revised to be suitable for publication in *eLife*.

1) One of the most significant features of this manuscript was the use of a novel behavioral paradigm. This was creative and the authors should be commended for developing this paradigm with a key control in place – the no-shock control group. However, the behavioral performance for this group were not shown. This is critically important in the readers' interpretation of the results. It is likely that the behavior within the no-shock control group would be different – if so, this could change the interpretation of the punishment-induced changes in theta/firing rate as either causing the behavioral change, resulting from a behavioral change, or signaling the perception of a punishing or potentially threatening context. I would like to see the behavior in both the experimental and control groups analyzed and displayed in greater detail to allow readers to appreciate 1) the variability within groups (animal-by-animal basis), 2) the variability between groups (was the variance greater for the punished group?), 3) the variability within-subjects (trial-to-trial variability). This should be done for both the shock and no-shock groups. I would also like to see how no-shock animals would behave under the influence of diazepam (presumably they would either have little to no change or even an increase in RT). Note that these comments can be addressed without further electrophysiology, and largely by reanalyzing data that the authors have already collected.

2) Along these lines, it would be useful to see a finer-grained time course across the session. Is the behavior stable within each block? Or is there a ramping across the session? Alternatively, do different animals respond differently at the beginning of each block?

3) The non-shock control was nice to see and convincingly controls for confounding variables such as time across the session and satiety. However, it was unclear if these are the same animals and in what order they were run, and whether these animals were naïve to the context and shock experience – please provide this information. Ideally, the experimental paradigm would have included an additional "no shock" block following the 3rd (10% shock) block within the same recording session to allow us to see how the same neurons would respond. While it raises the question of whether this is reversible and bidirectionally flexible, I understand that progressing nonlinearly (0%10%6%) may potentially be difficult to interpret since DA and/or the effects of DA receptor signaling could persist. But the time course of returning to baseline would be valuable to see.

4) The authors correctly described their groups as "putative VTA DA" in one figure but then proceed to refer to these neurons as "VTA DA neurons" throughout the rest of the manuscript. Without further support (using a genetic indicator for dopamine in addition to electrophysiological characteristics), they must refer to these neurons as putative throughout the manuscript (at least in the figures). At this point in the field, it is common to see both genetically-identified (using a histologically-verified transgenic rodent line, and sometimes even multiple lines) DA neurons as well as studies using electrophysiologically-identified DA neurons which has been shown to be a less reliable method for identification. As such, clear indication towards the strength of these claims should be reflected in each figure and legend so as not to mislead readers.

5) Precise mapping of the electrode locations correlated with any variability within the data should be displayed – although the authors do provide a schematic indicating the location of their recordings, these schematics are not plotted in relation to any differences in the neurons sampled per animal. A scatterplot would help. VTA DA innervation of PFC neurons is not even through the layers, and it would be valuable to know whether they are recording from putative VTA DA neurons that are in the medial VTA (and therefore more likely to project to the PFC) or the lateral VTA (more likely to project to the NAc).

6) Figure 2-score traces for putative VTA-DA and VTA-other neurons appears to rise prior to the cue onset. How is this explained?

7) Although the data suggest VTA→mPFC directional synchrony, perhaps driven by VTA-DA neurons. The direct relationship between VTA DA neurons and PFC neural activity is unclear. Is this a result of a direct connection? Is this change even DA dependent? Local DA manipulations (i.e., pharmacology or optical manipulations) may further clarify the nature of this relationship. If not, these limitations of the study should be directly discussed in the main text.

---

## [Author Response]

Essential revisions:The reviewers were in agreement that there is a major question over whether the rats understand the contingency between nose poking and punishment given the low probability and modest experience they have or whether the effects of punishment are more of a state dependent effect. In other words, would the same effect have been achieved by delivering increasing numbers of non-contingent shocks in the blocks? The reviews mention several ways to address this analytically in the current data or by running a behavioral group and/or recording group with non-contingent shock. If it cannot be clearly established that the current results reflect contingency and/or the authors do not want to run more subjects, then it might be acceptable to refrain from making strong claims about this in the Introduction and Results, and then discuss the possibilities in the Discussion. The current design is obviously appropriate to model predator risks during foraging anyway, as predators are not contingent on picking a berry necessarily. But in this case, the reviewers want to emphasize that the changes to the way this is described in the Introduction and Results must be substantive, removing up front claims that there is an action-punishment contingency. It was also felt that a finer grained analysis of individual rats' behavior and neural activity might be useful in addressing this question and more generally.

This is a valid concern; it was clearly an oversight not to include more behavioral data in the original manuscript. We have made the following text revisions and addition of new data to address this major concern:

1) We have removed up front claims on acquisition of action-punishment “contingency.” In fact, the word “contingency” has been removed from most of the text including the title. Instead, we refer to the primary manipulation of our task as punishment risk. We have also added a discussion of this concern in the subsection “VTA and mPFC Neurons represent risk of punishment on different timescales”. For us, the overarching scientific question of the study was how risk of punishment lurking in the background of reward-seeking behavior is represented by the VTA-mPFC neural ensembles. Thus, this change in wording provides a better representation of the work.

2) As requested by reviewers, we provide additional data to reveal that risk of punishment increases trial-to-trial variability in reaction time in individual animals (Figure 1—figure supplement 1). We also include new data to show that neural population trajectories of simultaneously recorded units significantly correlated with the trial-to-trial variation of reaction times. These new analyses suggest that the VTA and mPFC ensembles track the behavioral alteration as a function of punishment risk on a trial-by-trial basis (Figure 6, Figure 6—figure supplement 1–Figure 6—figure supplement 2).

3) The current task design does have the punishment contingent on the action albeit at low probabilities (0.06 or 0.1). To support this point, we have added a video showing the stereotypical pattern of action in the face of punishment risk as a qualitative evidence for the contingency representation. This video shows that animals are timid/cautious at the time of instrumental action and make many “incomplete” nose pokes, which is never the case for nose pokes to the food trough. The latter is mechanically the same action to which punishment was never contingent on. Given this, we discuss possibilities that behavioral and neural correlates of punishment risk may be due both to state dependent effects and partial learning of action-punishment contingency.

Other important, but less essential points are outlined below.Title: According to eLife guidelines, I believe it needs to mention the model species.Reviewer #1:[…] For me the overriding question is whether the rats understand the contingency between their behavior and the punishment or whether the behavior and resultant neural changes reflect a change in anxiety state independent of contingency. This is an issue, because the probability of punishment is extremely low given the number of trials and amount of training. This is not inappropriate, since the authors predicate the significance of their model on foraging and the example that reward seeking often involves risk. In this context, I think it is not critical that the risk be contingent on the action. It is enough that it is contingent on the rats' presence in the block or the environment. So one way to address this is to change the language in the paper to reflect this distinction, deal with this issue forthrightly in the Discussion, etc. Indeed it is a bit unclear whether the authors really mean to make the strong argument one way or the other.

We agree that this was a major issue – please see our general response above as well as the new text in Discussion subsection “VTA and mPFC Neurons represent risk of punishment on different timescales”. It may also be helpful to consider our response to reviewer #3 point #2.

Reviewer #2:[…] What is the question the manuscript is asking and attempting to answer? This isn't clear.A substantial problem lies in the experimental design and the reference to the punishment blocks as action-punishment contingency. The wording suggests that the manuscript is explicitly examining an action followed by an aversive event. However, there is no evidence that the behaviour and neural changes seen across blocks are due to a 'contingency'. Given the few punishment trials (3/50 in block 2 and 5/50 in block 3), it is possible that random shock delivery could yield the same behavioural and neural data. Unfortunately, in the absence of these data, the conclusions would need to be altered. I feel it would be better to stick to more appropriate wording that describes the changes in the blocks as increased punishment probability on the background of rewarded actions with no claims whether the punishment needs to be contingent on the action. As a side note, the VTA DA signal around the action may represent an altered reward value as a result of shock delivery within a block (which is different to representing an action-punishment contingency).

Please see our general response above – the scientific question that we were interested in answering was how risk of punishment/aversive outcome during reward-seeking behavior is represented by the VTA-mPFC neural ensembles.

What was the temporal firing of VTA non-DA neurons relative to VTA DA neurons? Is there an interaction between those two types of VTA neurons in the present context?

We had looked at cross-correlation of putative DA and non-DA cell pairs within the peri-action epoch, and found a small fraction of pairs with significant excitatory or inhibitory interactions that might be potentially monosynaptic (cell pairs with their significant cross-correlogram peaks < 10 ms). We decided against including this data in the original (and revised) manuscript because the number of cell pairs with meaningful cross-correlations was low. The small sample size would make it difficult for us to validly quantify how the neuronal pairwise interaction alters across blocks. We believe the interaction and potential alteration by punishment risk is a critical issue that can be better addressed in a future study with larger sampling of cell pairs (using a high-density silicon probe) ideally with a better labelling of cell identities.

It is hard to know if the rats knew about the upcoming shock probabilities in Block 2 and 3 or if they learned about those on the fly (i.e. state vs. learning effects). Data from Block 2 would be particularly telling in this regard and I think more data from that block should be added to clarify the question. For example, behavioural and neural data from the start of block 2 prior to first shock delivery would provide evidence whether the rats entered block 2 knowing about the increased shock probability (as argued in the manuscript) or behaved similarly to block 1 and altered their behaviour following the first shock presentation. With regard to the third block, the delivery of shock in the second block would likely have been a predictor for the upcoming increase in shock probability. This too can be inferred from the start of block 3 prior to first shock delivery in the form of increased RT.

To address this concern, we show additional data that demonstrates that RT during early trials in Blocks 2 and 3, *before* animals experienced the 1^st^ shock, increased as compared to block 1. This increase was subtle but significant in block 2 but quite robust in block 3 (Figure 1—figure supplement 1). These results suggest that animals were aware of the upcoming increase in punishment risk at least to some extent due to learning effects. On the other hand, we show that RT varies significantly across trials as a function of the trial lag from the previous shock trial in block 2 and 3. This variation in RT is suggestive of state effects on animals’ behavior. Considering these, we do not believe nor claim that behavioral changes we observed are due exclusively to either learning or state effect. Our data rather suggest that both learning and state effects contribute to behavioral changes.

Reviewer #3:Overall, I am enthusiastic about this study – though I do have some suggestions for improving the manuscript. This work is of interest, adding to the growing literature of the role of VTA-DA neurons and its relationship to the mPFC and aversively-motivated behavior. It certainly has many merits including the novel behavioral paradigm and the simultaneous recordings in PFC and VTA. Perhaps the most intriguing finding, is that punishment contingencies increase the number of putative VTA-DA units that are excited at the time of action (nose-poke). It is unfortunate that the manuscript cannot shed more light on the nature of the precise connections between the VTA and mPFC or the causal relationship between their findings and behavioral output. That said, I believe this manuscript could be revised to be suitable for publication in eLife.1) One of the most significant features of this manuscript was the use of a novel behavioral paradigm. This was creative and the authors should be commended for developing this paradigm with a key control in place – the no-shock control group. However, the behavioral performance for this group were not shown. This is critically important in the readers' interpretation of the results. It is likely that the behavior within the no-shock control group would be different – if so, this could change the interpretation of the punishment-induced changes in theta/firing rate as either causing the behavioral change, resulting from a behavioral change, or signaling the perception of a punishing or potentially threatening context.

We agree with the reviewer that further analysis of no-shock control behavioral data and behavioral data in general might be critical especially given our use of a novel behavioral paradigm. As the reviewer correctly points out below, we had not included any data showing behavioral variability across trials on animal-by-animal basis. We have added this information in the Figure 1—figure supplement 1 both for behavioral data from experimental and no-shock control sessions. We hope this additional information may help readers to better interpret animals’ performance in the task and neural correlates thereof.

I would like to see the behavior in both the experimental and control groups analyzed and displayed in greater detail to allow readers to appreciate 1) the variability within groups (animal-by-animal basis), 2) the variability between groups (was the variance greater for the punished group?), 3) the variability within-subjects (trial-to-trial variability). This should be done for both the shock and no-shock groups. I would also like to see how no-shock animals would behave under the influence of diazepam (presumably they would either have little to no change or even an increase in RT). Note that these comments can be addressed without further electrophysiology, and largely by reanalyzing data that the authors have already collected.2) Along these lines, it would be useful to see a finer-grained time course across the session. Is the behavior stable within each block? Or is there a ramping across the session? Alternatively, do different animals respond differently at the beginning of each block?

To address these questions, we did the following additional analyses:

1) We analyzed the behavioral data to reveal the extent to which RT varied across trials on an animal-by-animal basis (Figure 1—figure supplement 1). We show that the variability in RT increases as a function of punishment risk across blocks, and this increase in behavioral variability is associated with shock episodes in risky blocks, since the RT varied as a function of the trial lag from the previous shock (Figure 1).

2) We ran an additional population-level analysis which extracts trial-by-trial neural population trajectories within the population state space capturing the neuronal activity co-modulation structure of a given neuronal population (Materials and methods). We show that the trial-to-trial variability in RT is significantly correlated with trial-by-trial neural population activity measured in the population state space. This supports the notion that mPFC and VTA neuronal populations track the trial-by-trial RT variation in the face of punishment risk (Figure 6, Figure 6—figure supplement 1).

3) We conducted the behavioral variability and neural population trajectory analyses on the no-shock control data (Figure 1—figure supplement 1, Figure 6—figure supplement 1–Figure 6—figure supplement 2), and show that the increase in behavioral variability and the trial-by-trial behavioral and neural correlation are unique phenomena arising in the face of punishment risk.

3) The non-shock control was nice to see and convincingly controls for confounding variables such as time across the session and satiety. However, it was unclear if these are the same animals and in what order they were run, and whether these animals were naïve to the context and shock experience – please provide this information. Ideally, the experimental paradigm would have included an additional "no shock" block following the 3rd (10% shock) block within the same recording session to allow us to see how the same neurons would respond. While it raises the question of whether this is reversible and bidirectionally flexible, I understand that progressing nonlinearly (0%10%6%) may potentially be difficult to interpret since DA and/or the effects of DA receptor signaling could persist. But the time course of returning to baseline would be valuable to see.

We clarified the order of the experiments in Materials and methods (subsection “An instrumental task with varying punishment risk”). Same animals underwent the no-shock control session first before they received a shock (naive), which is also clarified (see the aforementioned subsection, and in the Figure 1 legend). Perhaps a bit out of the scope of the current study, but we agree that it is an important subject of future research to reveal how the cortical and midbrain neurons represent the probabilistic distributions and bidirectional changes in punishment risk.

4) The authors correctly described their groups as "putative VTA DA" in one figure but then proceed to refer to these neurons as "VTA DA neurons" throughout the rest of the manuscript. Without further support (using a genetic indicator for dopamine in addition to electrophysiological characteristics), they must refer to these neurons as putative throughout the manuscript (at least in the figures). At this point in the field, it is common to see both genetically-identified (using a histologically-verified transgenic rodent line, and sometimes even multiple lines) DA neurons as well as studies using electrophysiologically-identified DA neurons which has been shown to be a less reliable method for identification. As such, clear indication towards the strength of these claims should be reflected in each figure and legend so as not to mislead readers.

As suggested, we have labeled these cells as putative DA and putative non-DA units in all the figures, legends, and in many places within the body of the manuscript.

5) Precise mapping of the electrode locations correlated with any variability within the data should be displayed – although the authors do provide a schematic indicating the location of their recordings, these schematics are not plotted in relation to any differences in the neurons sampled per animal. A scatterplot would help. VTA DA innervation of PFC neurons is not even through the layers, and it would be valuable to know whether they are recording from putative VTA DA neurons that are in the medial VTA (and therefore more likely to project to the PFC) or the lateral VTA (more likely to project to the NAc).

To address this, we have looked at whether some of the VTA neural correlates vary systematically as a function of the medial-lateral placement of the electrodes. For example, we have plotted individual VTA putative DA neuronal PLVs across the ML axis of the electrode placement, which exhibits no systematical ML gradient (Author response image 1). We also have confirmed that the LFP power did not differ across different ML placements. Based on these, we have decided against claiming any spatial components within our VTA neural correlates.

6) Figure 2-score traces for putative VTA-DA and VTA-other neurons appears to rise prior to the cue onset. How is this explained?

The number of spikes were counted in 50-ms bins, and the binned spike counts were smoothed across adjacent time bins when estimating the normalized peri-event firing rates. We added further details in the Materials and methods (subsection “Neural data analysis”, second paragraph).

7) Although the data suggest VTA→mPFC directional synchrony, perhaps driven by VTA-DA neurons. The direct relationship between VTA DA neurons and PFC neural activity is unclear. Is this a result of a direct connection? Is this change even DA dependent? Local DA manipulations (i.e., pharmacology or optical manipulations) may further clarify the nature of this relationship. If not, these limitations of the study should be directly discussed in the main text.

We discuss these limitations and have added a statement acknowledging that more work is required to establish causation, and whether the VTA->mPFC drive of theta oscillation is a result of direct vs. multi-synaptic connections.